# Generalized Gaussian Temporal Difference Error for Uncertainty-aware Reinforcement Learning

## Abstract

Conventional uncertainty-aware temporal difference (TD) learning methods often rely on simplistic assumptions, typically including a zero-mean Gaussian distribution for TD errors. Such oversimplification can lead to inaccurate error representations and compromised uncertainty estimation. In this paper, we introduce a novel framework for generalized Gaussian error modeling in deep reinforcement learning, applicable to both discrete and continuous control settings. Our framework enhances the flexibility of error distribution modeling by incorporating additional higher-order moment, particularly kurtosis, thereby improving the estimation and mitigation of data-dependent noise, i.e., aleatoric uncertainty. We examine the influence of the shape parameter of the generalized Gaussian distribution (GGD) on aleatoric uncertainty and provide a closed-form expression that demonstrates an inverse relationship between uncertainty and the shape parameter. Additionally, we propose a theoretically grounded weighting scheme to fully leverage the GGD. To address epistemic uncertainty, we enhance the batch inverse variance weighting by incorporating bias reduction and kurtosis considerations, resulting in improved robustness. Extensive experimental evaluations using policy gradient algorithms demonstrate the consistent efficacy of our method, showcasing significant performance improvements.

## 1 Introduction

Deep reinforcement learning (RL) has demonstrated promising potential across various real-world applications, e.g., finance (Moody & Saffell, 1998; Byun et al., 2023; Sun et al., 2023), and autonomous driving (Kahn et al., 2017; Emamifar & Ghoreishi, 2023; Hoel et al., 2023). One critical avenue for improving the performance and robustness of RL agents in these complex, high-dimensional environments is the quantification and integration of *uncertainty* associated with the decisions made by agents or the environment (Lockwood & Si, 2022). Effective management of uncertainty promotes the agents to make more informed decisions leading to enhanced sample efficiency in RL context, which is particularly beneficial in unseen or ambiguous situations.

Temporal difference (TD) learning is a fundamental component of many RL algorithms, facilitating value function estimation and policy derivation through iterative updates (Sutton, 1988). Traditionally, these TD updates are typically grounded in $L_2$ loss, corresponding to maximum likelihood estimation (MLE) under the assumption of Gaussian error. Such simplification may be overly restrictive, especially considering the noisy nature of TD errors, which are based on constantly changing estimates of value functions and policies. This assumption compromises sample efficiency, necessitating the incorporation of additional distributional parameters for flexible but computationally efficient TD error modeling.

In statistics and probability theory, distributions are typically characterized by their central tendency, variability, and shape (DeCarlo, 1997; Milton et al., 2017). Traditional deep RL methods, however, effectively exploit on the variance of the error distribution through the scale parameter, yet they often disregard its *shape*. This oversight hinders these methods from fully capturing the true underlying uncertainty. The kurtosis, the scale-independent moment, does significantly influence both infer-

ential and descriptive statistics (Balanda & MacGillivray, 1988), and the reliability of uncertainty estimation.

Therefore, it is essential to incorporate the shape of the error distribution into TD learning to better reflect uncertainties present in RL environments, enabling more robust and reliable decision-making processes in dynamic and complex scenarios.

A notable enhancement to the normality hypothesis is the use of the *generalized Gaussian distribution* (GGD), also known as the generalized error distribution or exponential power distribution. This flexible family of symmetric distributions, as depicted in Figure 1, encompasses a wide range of classical distributions, including Gaussian, Laplacian, and uniform distributions, all adjustable via a shape parameter (Box & Tiao, 2011). This specific parameter allows for fine-tuning the distribution to match the characteristics of TD error distribution, providing a more reliable representation of uncertainty.

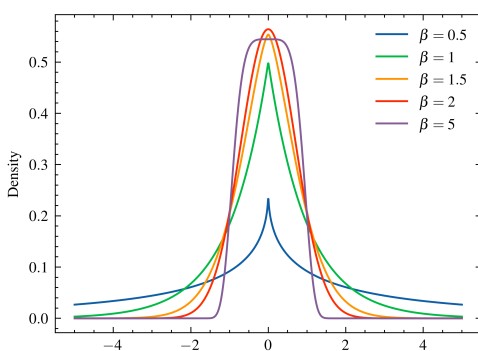

Figure 1: Generalized Gaussian distribution.

**Contributions** Our primary contribution is the introduction of a novel framework of generalized Gaussian error modeling in deep RL, enabling robust training methodologies by incorporating the distribution's shape. This approach addresses both data-dependent noise, i.e., aleatoric uncertainty, and the variance of model estimates, i.e., epistemic uncertainty, ultimately enhancing model stability and performance.

The key contributions of our work are as follows:

1. **Empirical investigations** (Section 3.1.1): We conduct empirical investigations of TD error distributions, revealing substantial deviations from the Gaussian distribution, particularly in terms of tailedness. These findings underscore the limitations of conventional Gaussian assumptions.

2. **Theoretical exploration** (Section 3.1.2): Building on empirical insights, we explore the theoretical suitability of modeling TD errors with a GGD. Theorem 1 demonstrates the effectiveness and well-definedness of the proposed method under leptokurtic error distributions, characterized by $\beta \in (0, 2]$. Our experimental results suggest that the estimates of $\beta$ mostly converge within this range, aligning with the empirical findings.

3. **Aleatoric uncertainty mitigation** (Section 3.1): We investigate the implications of the distribution shape on the estimation and mitigation of aleatoric uncertainty. GGD error modeling enables the quantification of aleatoric uncertainty in a closed form, indicating a negative relation to the shape parameter $\beta$ on an exponential scale, with a constant scale parameter $\alpha$. We also leverage the second-order stochastic dominance of GGD to weight error terms proportional to $\beta$, enhancing the model's robustness to heteroscedastic aleatoric noise by focusing on less spread-out samples.

4. **Epistemic uncertainty mitigation** (Section 3.2): We introduce the batch inverse error variance weighting scheme, adapted from the batch inverse variance scheme (Mai et al., 2022), to account for both variance and kurtosis of the estimation error distribution. This scheme down-weights high-variance samples to prevent noisy data and improves model robustness by focusing on reliable error estimates.

5. **Experimental evaluations** (Section 4): We conduct extensive experimental evaluations using policy gradient algorithms, demonstrating the consistent efficacy of our method and significant performance enhancements.

## 2 BACKGROUND

We consider a Markov decision process (MDP) governed by state transition probability $\mathcal{P}(s_{t+1}|s_t, a_t)$, with $s_t \in \mathcal{S}$ and $a_t \in \mathcal{A}$ represent the state and action at step $t$, respectively (Sut-

ton & Barto, 2018). Within this framework, an agent interacts with the environment via a policy $\pi(a_t|s_t)$, leading to the acquisition of rewards $r(s_t, a_t) \sim \mathcal{R}(s_t, a_t)$.

Numerous model-free deep RL algorithms leverage TD updates for value function approximation (Mnih et al., 2015; 2016; Schulman et al., 2017; Fujimoto et al., 2018; Haarnoja et al., 2018). In these methods, neural networks parameterized by $\theta$ are trained to approximate the state-action value $Q(s_t, a_t)$ by minimizing the error between the target and predicted value:

$$\delta(s_t, a_t; \theta) = T(s_t, a_t; \theta) - Q(s_t, a_t; \theta), \tag{1}$$

where the target is computed according to Bellman's equation:

$$T(s_t, a_t; \theta) = r(s_t, a_t) + \gamma Q(s_{t+1}, a_{t+1}; \theta). \tag{2}$$

Typically, TD updates involve minimizing the mean squared error (MSE) loss $\text{MSE}_\theta = \mathbb{E}[(T(s_t, a_t; \theta) - Q(s_t, a_t; \theta))^2]$ through stochastic gradient descent. This minimization implicitly presumes that errors conform to a Gaussian distribution with zero-mean, consistent with the principles of MLE.

For clarity, we henceforth omit the learnable parameter and adopt subscript notation for function arguments, e.g., $\delta_t = T_t - Q_t$.

## 2.1 UNCERTAINTY

Uncertainty in neural networks is commonly decomposed into two sources: aleatoric and epistemic (Der Kiureghian & Ditlevsen, 2009; Kendall & Gal, 2017; Depeweg et al., 2018; Valdenegro-Toro & Mori, 2022). Epistemic uncertainty arises from limitations within the neural network and can potentially be reduced through further learning or model improvements. In contrast, aleatoric uncertainty stems from the inherent stochasticity of the environment or the dynamics of the agent-environment interactions and is fundamentally irreducible.

This distinction is crucial in the context of RL, where areas with high epistemic uncertainty need to be explored, whereas exploring areas with high aleatoric uncertainty may lead to ineffective training, since the agent might have adequate knowledge but insufficient information for decisive actions. Quantifying aleatoric uncertainty is known to facilitate learning dynamics of stochastic processes and enables risk-sensitive decision making (Dabney et al., 2018; Vlastelica et al., 2021; Seitzer et al., 2022).

To address aleatoric uncertainty, variance networks are frequently employed. These networks, denoted as $Q^\sigma$, are used in conjunction with value approximation networks $Q^\mu$ (Bishop, 1994; Nix & Weigend, 1994; Kendall & Gal, 2017; Lakshminarayanan et al., 2017; Mai et al., 2022; Mavor-Parker et al., 2022). The estimated variance from $Q^\sigma$ is utilized for heteroscedastic Gaussian error modeling, which captures aleatoric uncertainty (Seitzer et al., 2022). Specifically, the TD errors are modeled under a Gaussian distribution, i.e., $\delta_t \sim \mathcal{N}(Q_t^\mu, Q_t^{\sigma 2})$, and the network minimizes the Gaussian negative log-likelihood (NLL). This formulation penalizes errors proportionally to their predicted variance. Larger $Q_t^\sigma$ values, indicating greater aleatoric uncertainty, reduce the penalty for large errors, effectively addressing their impact.

To mitigate epistemic uncertainty, the batch inverse variance (BIV) regularization method, as proposed in Mai et al. (2022), is applied. This approach scales error contributions inversely to their variance, ensuring that noisy samples contribute less to the gradient. The BIV weight is defined as $\omega_t^{\text{BIV}} = 1/(\gamma^2 \mathbb{V}[Q_t^\mu] + \xi)$, where empirical variance $\mathbb{V}[Q_t^\mu]$ is the empirical variance of the ensembled value heads $Q_t^\mu$, and $\xi$ is either a hyperparameter or numerically computed to ensure a sufficient effective batch size. Integrating this into the overall loss results in the BIV loss:

$$\mathcal{L}_{\text{BIV}} = \left( \frac{\omega_t^{\text{BIV}}}{\sum_\tau \omega_\tau^{\text{BIV}}} \delta_t^2 \right).$$

The complete loss function, combining aleatoric and epistemic uncertainty terms, is given by:

$$\mathcal{L} = \mathcal{L}_{\text{GD-NLL}} + \lambda \mathcal{L}_{\text{BIV}}$$

$$= \sum_t \left( (\delta_t/Q_t^\sigma)^2 + \log Q_t^{\sigma 2} \right) + \lambda \left( \frac{\omega_t^{\text{BIV}}}{\sum_\tau \omega_\tau^{\text{BIV}}} \delta_t^2 \right). \tag{3}$$

Here, $\lambda$ is a regularizing temperature. This approach balances uncertainty quantification, leveraging variance networks for aleatoric modeling and BIV regularization for robust handling of epistemic uncertainty. The use of empirical variance with Bessel's correction ensures robust variance estimates, especially in small sample scenarios, e.g., ensemble sizes of five in the official implementation. Note that the variance among the ensembled critics is employed to estimate epistemic uncertainty, with the variance estimator serving as an empirical measure of aleatoric uncertainty.

## 2.2 TAILEDNESS

While mainstream machine learning literature often prioritizes on capturing central tendencies, the significance of extreme events residing in the tails for enhancing performance and gaining a deeper understanding of learning dynamics cannot be overlooked. This is especially relevant for MLE base on the normality assumption, which is commonly applied in variance network frameworks. Focusing solely on averages or even deviations is proven to be inadequate in the presence of outlier samples (David, 1979; Gather & Kale, 1988).

For instance, consider the impact of non-normal samples on the estimate of the variance, as described in Proposition 1 with proof presented in Appendix B.1.

**Proposition 1** (Biased variance estimator (Yuan et al., 2005))**.** *Let $X_1, X_2, ..., X_n$ be a sequence of independent, non-normally distributed random variables from a population $X$ with mean $\mu$, variance $\sigma^2$, and kurtosis $\kappa$. Then, with the MLE estimators under normality assumption, i.e., $\hat{\mu} = \sum_{i=1}^{n} X_i/n$ for mean and $\hat{\sigma}^2 = \sum_{i=1}^{n}(X_i - \hat{\mu})^2/n$ for variance, the variance estimator $\hat{\sigma}^2$ exhibits bias. Specifically, it will be negatively biased when $\kappa > 0$ and positively biased when $\kappa < 0$.*

Proposition 1 elucidates that the standard error of estimated TD error variance is a function of kurtosis $\kappa$. With heavy-tail distributions, the standard error of variance estimates, through networks trained by Gaussian NLL (Nix & Weigend, 1994; Bishop, 1994), may also be underestimated, highlighting the influence of kurtosis on variance estimation. Furthermore, it is shown that the likelihood ratio statistics for variance estimator depends on kurtosis even for large $n$ (Yuan et al., 2005). This emphasizes the necessity of accounting for tailedness to derive robust variance estimates confidently.

*Remark* 1 (Varietal variance estimator (Burch, 2014))*.* The dependence of the standard error of the variance estimator on kurtosis impacts both the bias and variance of variance estimation. Specifically, when the kurtosis of the underlying distribution exceeds zero, assuming normality tends to result in an underestimation of the confidence interval of variance estimation.

**Gumbel error modeling** Recent applications of Gumbel distribution closely related to TD learning have emerged for estimating the maximum $Q$ value in the Bellman update process (Garg et al., 2022; Hui et al., 2023). These approaches capitalize on the foundation of the extreme value theorem, which states that maximal values drawn from any exponential-tailed distribution follow the Gumbel distribution (Fisher & Tippett, 1928; Mood, 1950). Compared to conventional distributional RL algorithms employing Gaussian mixture or quantile regression (Dabney et al., 2018; Shahriari et al., 2022), this approach showcases superior control performance. However, it has been observed that while Gumbel modeling initially aligns with the error distribution propagated through the chain of max operations, its Gumbel-like attribute may diminish over the course of training (Garg et al., 2022). We instead propose a novel approach utilizing GGD, which offers flexibility in expressing the tail behavior of diverse distributions. This method is adaptable to wider range of MDPs, even those without max operators.

## 3 METHODS

Our approach introduces enhancements to the loss function derived from Equation (3), incorporating tailedness into both loss attenuation (Section 3.1) and regularization terms (Section 3.2):

$$\mathcal{L} = \mathcal{L}_{\text{GGD-NLL}}^{\text{RA}} + \lambda \mathcal{L}_{\text{BIEV}}$$

$$= \sum_t \frac{\omega_t^{\text{RA}}}{\sum_\tau \omega_\tau^{\text{RA}}} \left( \left( |\delta_t| / Q_t^\alpha \right)^{Q_t^\beta} - \log Q_t^\beta / Q_t^\alpha + \log \Gamma(1/Q_t^\beta) \right) + \lambda \left( \frac{\omega_t^{\text{BIEV}}}{\sum_\tau \omega_\tau^{\text{BIEV}}} \delta_t^2 \right), \quad (4)$$

where $Q^\alpha$ and $Q^\beta$ represent the alpha and beta networks, respectively. Here, risk-averse weights $\omega_t^{\text{RA}} = Q_t^\beta$, and batch inverse error variance (BIEV) weights $\omega_t^{\text{BIEV}} = 1/(\mathbb{V}[\delta_t] + \xi)$. These dual

loss terms effectively capture aleatoric and epistemic uncertainties, respectively. The subsequent subsections provide a detailed rationale for this modification.

## 3.1 GENERALIZED GAUSSIAN ERROR MODELING

One simple yet promising approach to address non-normal heteroscedastic error distributions involves modeling the per-error distribution using a zero-mean symmetric GGD (Zeckhauser & Thompson, 1970; Chai et al., 2019; Giacalone, 2020; Upadhyay et al., 2021):

$$\delta \sim \text{GGD}(0, \alpha, \beta) = \frac{\beta}{2\alpha\Gamma(1/\beta)} e^{-(|\delta|/\alpha)^\beta}, \tag{5}$$

where $\alpha$ and $\beta$ represent the scale and shape parameter, respectively. This method not only allows for modeling each non-identical error by parameterizing the GGD with different $\alpha_t$ and $\beta_t$ at step $t$, but also offers a flexible parametric form that adapts across a spectrum of classical distributions from Gaussian to uniform as $\beta$ increases to infinity (Giller, 2005; Nadarajah, 2005; Novey et al., 2009; Dytso et al., 2018).

The shape parameter $\beta$ serves as a crucial structure characteristic, distinguishing underlying mechanisms. The kurtosis $\kappa$, commonly used to discern different distribution shapes, is solely a function of $\beta$ and is defined as Pearson's kurtosis minus three to emphasize the difference from Gaussian distribution (DeCarlo, 1997):

$$\kappa = \frac{\Gamma(5/\beta)\Gamma(1/\beta)}{\Gamma(3/\beta)^2} - 3. \tag{6}$$

This implies that distributions with $\beta < 2$ are leptokurtic, i.e., $\kappa > 0$, indicating a higher frequency of outlier errors compared to the normal error distribution. With only one additional parameter to characterize the distribution, GGD effectively expresses differences in tail behavior, a capability distributions dependent solely on location or scale parameters lack.

*Remark* 2. Despite the GGD having three parameters, we only employ $\beta$ estimation for GGD error modeling to minimize computational overhead, setting the scale parameter $\alpha$ to one. While this may slightly limit the expressivity of the error model, it significantly enhances training stability. The impact of omitting the $\alpha$ parameter is minimal, as $\alpha$ and $\beta$ are interdependent, e.g., variance $\sigma^2 = \alpha^2\Gamma(3/\beta)/\Gamma(1/\beta)$ (Dytso et al., 2018). Additionally, this simplification offers implementation advantages, requiring only a slight change in the loss function to migrate from variance networks, i.e., from Gaussian to GGD NLL.

### 3.1.1 EMPIRICAL EVIDENCE

Figure 2 presents empirical findings that reveal a significant deviation from Gaussian distribution in TD errors, evidenced by well-fitted GGDs and pronounced differences in the shape of distributions. This non-normality becomes particularly apparent when contrasting initial and final evaluations, suggesting an increasing prominence of tailedness throughout the training process.

We hypothesize that such departure from normality stems from the exploratory nature of agent behavior. During exploration, agents frequently encounter *unexpected* states and rewards, i.e., regions of the state-action space or reward function that are rarely visited. This leads to a higher frequency of outlier errors, leading to a broader spectrum of TD errors than those seen in purely exploitative scenarios. This increased variability likely contributes to the emergence of non-normal distributions, characterized by heavier tails.

Furthermore, the observed evolution in the tails of TD errors underscores the shifting interplay between aleatoric and epistemic uncertainties. As training advances, epistemic uncertainty typically diminishes, which inherently has Gaussian-like characteristics as it is measured by the variance of the ensembled estimates (Kendall & Gal, 2017). Aleatoric uncertainty, on the other hand, arises from irreducible noise inherent in the environment, e.g., stochasticity in rewards or transitions, and becomes more pronounced as the agent explores new states and actions. Such dynamics potentially result in non-normally distributed errors with heavier tails. Notably, the diminishing property of Gumbel-like attribute, discussed in Section 2.2, is also reflected in the evolving TD error distributions.

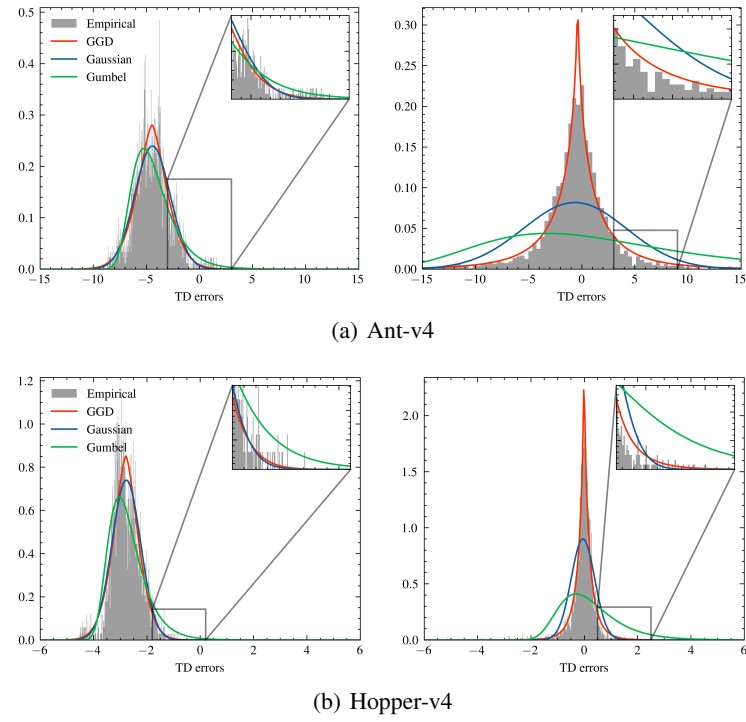

(a) Ant-v4

(b) Hopper-v4

Figure 2: TD error plots of SAC at the initial and final evaluations, arranged left to right, with fitted probability density functions (PDFs) using SciPy (Virtanen et al., 2020). Additional plots on other environments and for PPO are available in Appendix A.1.

### 3.1.2 THEORETICAL ANALYSIS

Given the empirical suitability of GGD for modeling TD errors, we conduct an in-depth theoretical examination of GGD. We begin by demonstrating the well-definedness of GGD regression, with a specific emphasis on the positive-definiteness of its PDF under certain conditions.

**Theorem 1** (Positive-definiteness (Bochner, 1937; Ushakov, 2011; Dytso et al., 2018)). *The NLL of GGD is well-defined for $\beta \in (0, 2]$.*

This theorem guarantees the well-definedness of the NLL of the GGD. It ensures that the PDF is guaranteed to be positive everywhere under highly probable conditions of $\beta$, thus affirming the suitability of the NLL as a loss function for a valid probability distribution. In fact, it is known that parameter estimation for GGD can also be numerically accomplished by minimizing the NLL function, with asymptotic normality, consistency, and efficiency of the estimates ensured under suitable regularity conditions (Agro, 1995). This positive-definiteness not only implies a theoretical property but also has practical implications in deep RL, where ensuring stability and convergence is crucial. The positive-definiteness of the PDF of GGD shown in the proof, elaborated in Appendix B.2, also assures that the function integrates to one.

The shape parameter $\beta$ also introduces a desirable property of *risk-aversion* to GGD, which can be mathematically formulated by stochastic dominance (Levy, 1992; Martin et al., 2020). Stochastic dominance, a concept assessing random variables via a stochastic order, reflects the shared preferences of rational decision-makers.

**Theorem 2** (Second-order stochastic dominance (Dytso et al., 2018)). *Consider two random variables $X_1 \sim GGD(0, \alpha, \beta_1)$ and $X_2 \sim GGD(0, \alpha, \beta_2)$, where $\alpha > 0$ and $\beta_1, \beta_2 > 0$. If $\beta_1 \leq \beta_2$, then $X_2$ exhibits second-order stochastic dominance over $X_1$. This dominance implies, for all $x$,*

$$\int_{-\infty}^{x} [F_{X_1}(t) - F_{X_2}(t)]\, dt \geq 0, \tag{7}$$

*where $F$ denotes the cumulative distribution function.*

The above theorem, with its proof detailed in Appendix B.3, suggests second-order stochastic dominance of TD errors. This dominance signifies a preference for risk-aversion, wherein the dominant variable $X_2$ exhibits greater predictability and maintains expectations that are equal to or higher than those of $X_1$ for all concave and non-decreasing functions (Osband & Van Roy, 2017). Formally, for such a function $u : \mathbb{R} \to \mathbb{R}$, we have $\mathbb{E}[u(X_2)] \geq \mathbb{E}[u(X_1)]$, with $\mathbb{E}$ denoting expectation.

The dominance relationship among GGD random variables, determined by the shape parameter $\beta$, reinforces the suitability of GGD for modeling errors in TD learning. Leveraging this characteristic, we propose a risk-averse weighting scheme $\omega_t^{\text{RA}} = Q_t^{\beta}$ by capitalizing on the tendency of GGD to learn from relatively less spread-out samples, thereby enhancing robustness to heteroscedastic noise.

*Remark* 3. The training of the critic is more directly influenced by aleatoric uncertainty, since only the critic loss is a direct function of the state, action, and reward, with the actor being downstream of the critic in uncertainty propagation. GGD error modeling enables us to quantify aleatoric uncertainty as a closed form, i.e., $\sigma^2 = \alpha^2 \Gamma(3/\beta)/\Gamma(1/\beta)$ (Upadhyay et al., 2021). Remarkably, aleatoric uncertainty exhibits a negative proportionality to the shape parameter $\beta$ on an exponential scale, with a constant scale parameter $\alpha = 1$ adopted in our implementation employing a beta head exclusively. Building on this, risk-averse weighting mitigates the negative impacts of noisy supervision by assigning higher weights to errors with lower aleatoric uncertainty for the loss attenuation term.

## 3.2 BATCH INVERSE ERROR VARIANCE REGULARIZATION

When estimating the uncertainty of target estimates, as employed in BIV weighting, potential bias can arise (Janz et al., 2019; Liang et al., 2022). Conversely, the bias of TD errors remains notably small with the assumption of constant value approximation bias (Flennerhag et al., 2020). Motivated by this, we propose the BIEV weighting:

$$\omega_t^{\text{BIEV}} = \frac{1}{\mathbb{V}[\delta_t]}, \tag{8}$$

incorporating the concept of *error variance* explicitly into BIV weighting. As mentioned in Section 2.1, the variance for BIEV regularization is estimated by the ensemble of critics, serving as an empirical measure of epistemic uncertainty.

Recent investigations have explored advancements in variance estimation, particularly through a constant multiplier, i.e., $s_\omega^2 = \omega(n-1)s^2$ (Kleffe, 1985; Wencheko & Chipoyera, 2009), where $s^2$ denotes sample variance, the MLE estimator of variance. Although non-standard weights $\omega \neq 1/(n-1)$ may introduce bias in variance estimation, the estimation of inverse variance remains biased due to Jensen's inequality, even with the use of the unbiased estimator $s^2$ (Walter et al., 2022). Consequently, our focus shifts to relative efficiency (RE), where we derive the MSE-best biased estimator (MBBE) in Proposition 2.

**Proposition 2** (MBBE of variance (Searls & Intarapanich, 1990; Wencheko & Chipoyera, 2009)). *Let $s_\omega^2 = \omega(n-1)s^2$ be the adjusted variance estimator, with the sample variance $s^2$ and the weight $\omega$ being a function of the sample size $n$ and the population kurtosis $\kappa$. Then, the estimator with the least MSE is given by:*

$$s_{\omega^*}^2 = \left( \frac{\kappa}{n} + \frac{n+1}{n-1} \right)^{-1} s^2. \tag{9}$$

*Additionally, MBBE of variance $s_{\omega^*}^2$ has consistent superior efficiency over the sample variance $s^2$, i.e.:*

$$RE_n = \frac{\mathbb{V}[s^2]}{\text{MSE}(s_{\omega^*}^2)} = 1 + \frac{\kappa}{n} + \frac{2}{n-1} > 1. \tag{10}$$

The reciprocal relationship between sample size and the impact of kurtosis on variance estimation is intuitive, especially for small samples where kurtosis is much more significant. Consequently, we advocate for the adoption of the MBBE in epistemic uncertainty estimation. It's worth noting that while the derivation of improved estimators presupposes known kurtosis, our method differs by utilizing estimated kurtosis.

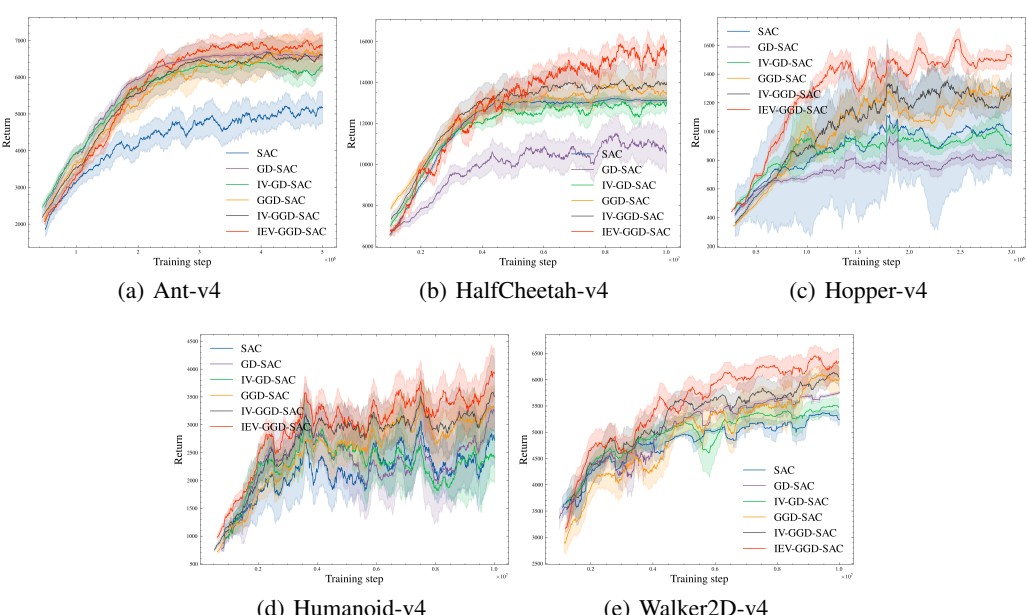

(a) Ant-v4        (b) HalfCheetah-v4        (c) Hopper-v4

(d) Humanoid-v4        (e) Walker2D-v4

Figure 3: Sample efficiency curves of SAC on MuJoCo environments, illustrating median return values averaged over ten random seeds. Shaded regions indicate the standard deviation. Prefixes denote applied techniques, e.g., 'GD-' for variance head, 'GGD-' for beta head, and 'IEV-' for BIEV regularization.

As BIV weighting, the BIEV weighting plays a crucial role in enhancing the robustness and efficiency TD updates. By normalizing the weight of each sample in a batch relative to the scale of its epistemic uncertainty compared to other samples, BIEV weighting ensures that the model appropriately accounts for the reliability of the estimate from each data point, resulting in robustness against inaccuracies in variance estimate calibration.

## 4 EXPERIMENTS

We conduct a comprehensive evaluation of our proposed method across well-established benchmarks, including MuJoCo (Todorov et al., 2012), and discrete control environments from Gymnasium (Towers et al., 2023). Notably, we augment the discrete control environments through the introduction of supplementary uniform action noise to enhance environmental fidelity.

To underscore the versatility and robustness of our approach, we deliberately choose baseline algorithms that cover a wide spectrum of RL paradigms. Specifically, we employ soft actor-critic (SAC) (Haarnoja et al., 2018), an off-policy $Q$-based policy gradient algorithm, and proximal policy optimization (PPO) (Schulman et al., 2017), an on-policy $V$-based method. We focus on adversaries limited to variance networks due to the use of separate target networks in previously mentioned Gumbel error modeling methods. This constraint is intended to focus on computationally efficient algorithms that only incorporate an additional layer, referred to as a *head*.

The algorithms are implemented using PyTorch (Paszke et al., 2019), within the Stable-Baselines3 framework (Raffin et al., 2021). We use default configurations from Stable-Baselines3, with adaptations limited to newly introduced hyperparameters. For both PPO and SAC, along with their variants, we employ five ensembled critics. The parameter $\xi$ from Equation (4) is computed with a minimum batch size of 16, and the regularizing temperature $\lambda$ is set to 0.1. Additional experimental details are provided in Appendix C.

The performance of SAC across different MuJoCo environments is presented in Figure 3. While the variance head degrades performance in certain scenarios, SAC variants employing the beta head consistently lead to better sample efficiency and asymptotic performance. Notable improvements are

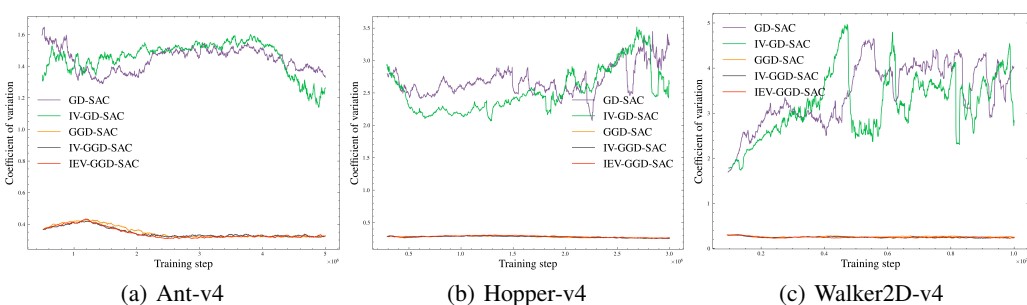

Figure 4: Coefficients of variation of parameter estimates for SAC variants. Results for other environments can be found in Appendix A.2.

observed in the HalfCheetah-v4 and Hopper-v4 environments, where the variance head substantially reduces sample efficiency. This suggests that the TD error distributions in these environments may exhibit heavy tails. The impact of BIEV regularization varies by environment but generally performs at least as well as BIV regularization.

Figure 4 shows the coefficients of variation, defined as the ratio of the standard deviation to the mean, i.e., $\sqrt{\mathbb{V}[X]}/\mathbb{E}[X]$, for the variance and $\beta$ estimates. This statistic demonstrates the scale-invariant volatility of parameter estimation, given that the scale of the estimated variance is significantly larger than that of the $\beta$ estimates. A lower coefficient of variation in $\beta$ estimation indicates greater stability compared to variance estimation.

The convergence of $\beta$ estimation is also more stable than variance estimation. This observation aligns with Remark 1, suggesting a potential underestimation of confidence intervals in variance estimation. Considering the susceptibility of variance estimates to extreme values, such underestimation introduces considerable uncertainty in parameter estimation. Our hypothesis regarding the escalating impact of extreme TD errors throughout training is consistent with this findings, as it exacerbates the challenge in variance estimation, leading to volatility or divergence of the variance head.

These findings support that utilizing the beta head results in lower and converging coefficients of variation in parameter estimation.

Figure 5 demonstrates the results of training PPO on MuJoCo and discrete control environments with additional noise. Remarkably, the incorporation of the beta head and BIEV regularization yields similar outcomes to those observed in SAC experiments. This indicates the efficacy of GGD error modeling in state value $V$-based TD learning as well.

We present comprehensive ablation studies in Appendix D, examining the efficacy of key components, including risk-averse weighting, regularizing temperature $\lambda$, BIEV regularization applied to SAC and its Gaussian variant, and the integration of the alpha head.

## 5 DISCUSSION

In this paper, we advocate for and substantiate the integration of GGD modeling for TD error analysis. Our main contribution is the introduction of a novel framework that enables robust training methodologies by leveraging the distribution's shape. This approach accounts for both data-dependent noise, i.e., aleatoric uncertainty, and the uncertainty of value estimation, i.e., epistemic uncertainty, ultimately enhancing the model's stability and accuracy.

**Further investigation** An imperative avenue for further investigation is the application of GGD within the context of maximum entropy RL. Similar to how the Gaussian distribution maximizes entropy with constraints up to the second moment, the GGD maximizes entropy subject to a constraint on the $p$-th absolute moment (MacKay, 2003). Exploring higher moments of the distribution could provide new insights into maximum entropy RL frameworks.

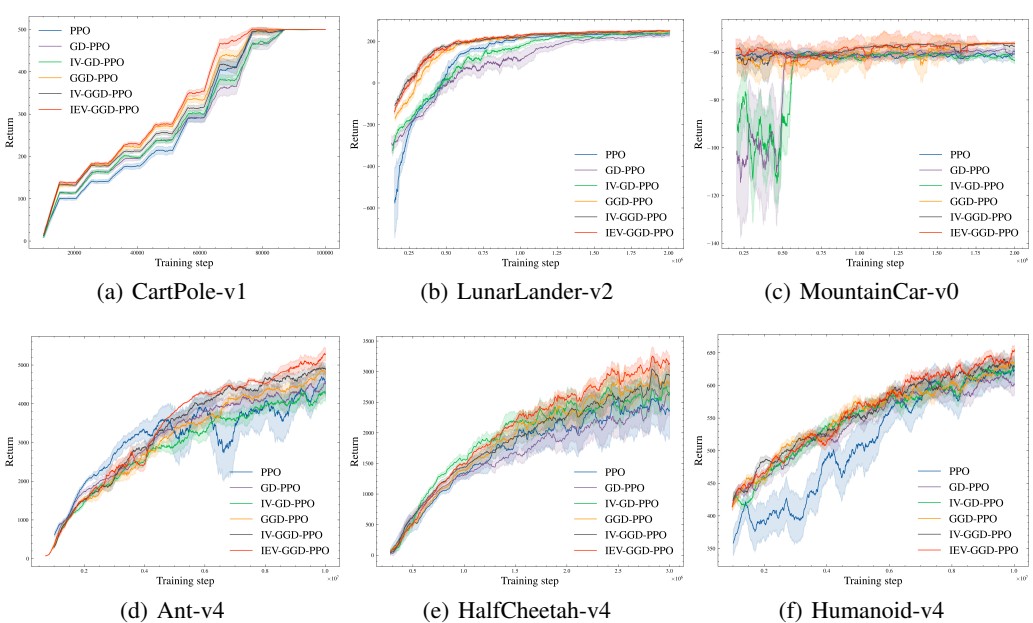

Figure 5: Sample efficiency curves of PPO on various control environments.

The absence of a comprehensive regret analysis in our current study also presents an opportunity for future work. Considering that aleatoric uncertainty in TD learning predominantly arises from reward dynamics, conducting a regret analysis on heavy-tailed TD error is warranted. This is particularly relevant as previous research has extensively studied regret in the context of heavy-tailed rewards.

While our experiments focus on policy gradient methods, the implications of TD error tailedness extend to the $Q$-learning family of algorithms. We provide empirical findings on this extension in Appendix E, which demonstrate the broader applicability of our approach. Additionally, exploring the generalized extreme value distribution could further enrich our understanding of tailedness phenomena due to its close association with extreme value theory.

**Relevant applications**    The implications of tailedness in TD error distributions extend into various domains, notably robust RL and risk-sensitive RL. The focus in robust RL lies on developing algorithms that are less sensitive to noise and outliers within the reward signal. Recognizing potential deviations from normality in TD error distributions is critical for designing such algorithms. Our work emphasizes the importance of considering non-normal error distributions, especially the tail behaviors, to enhance the robustness of RL algorithms.

Another significant direction is risk-sensitive RL, which seeks to assess and mitigate the risks associated with different action choices. In noisy and outlier-prone environments, capturing the risk profile using a Gaussian assumption for TD errors might be inadequate. By considering the GGD, which better models the tail behaviors of error distributions, we can develop more accurate and reliable risk-sensitive RL algorithms.

In summary, our exploration into GGD modeling of TD errors opens several promising research directions and applications, emphasizing the need to consider non-normal error distributions for enhancing the robustness and risk-sensitivity of RL algorithms.

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

## A    EXTENDED RESULTS

### A.1    TEMPORAL DIFFERENCE ERROR PLOTS

We present the distributions of TD errors sampled at the initial and final evaluation steps, depicted in Figures 2 and 6 for SAC, and Figure 7 for PPO, which highlights the heavy tailedness of TD errors and the tendency converge to heavy tail throughout training. This finding also emphasizes how aleatoric uncertainty affects their distribution, as elaborated in Section 3.1. Interestingly, both state-action values $Q$ and state values $V$ demonstrate similar characteristics in their TD error distributions.

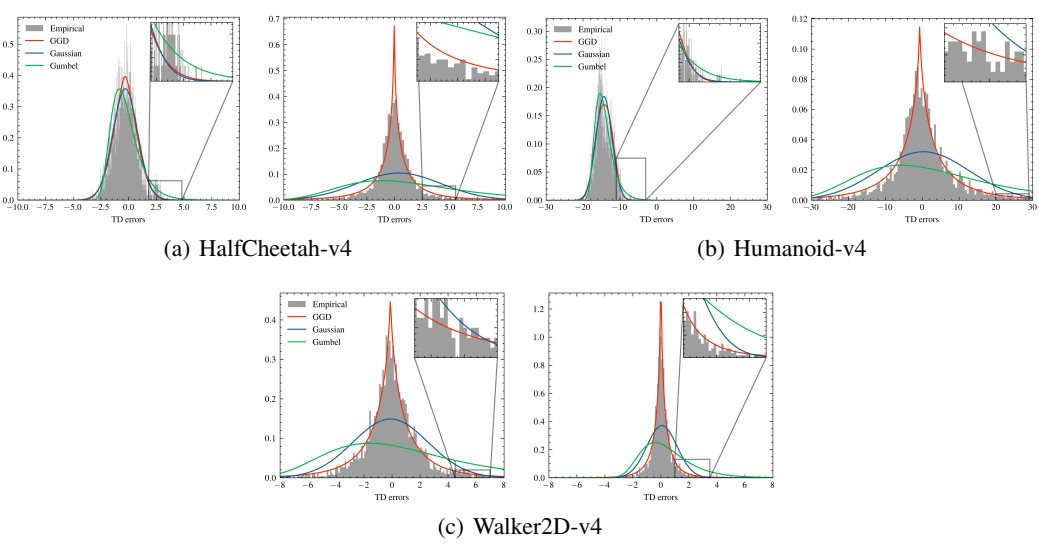

Figure 6:  TD error plots of SAC.

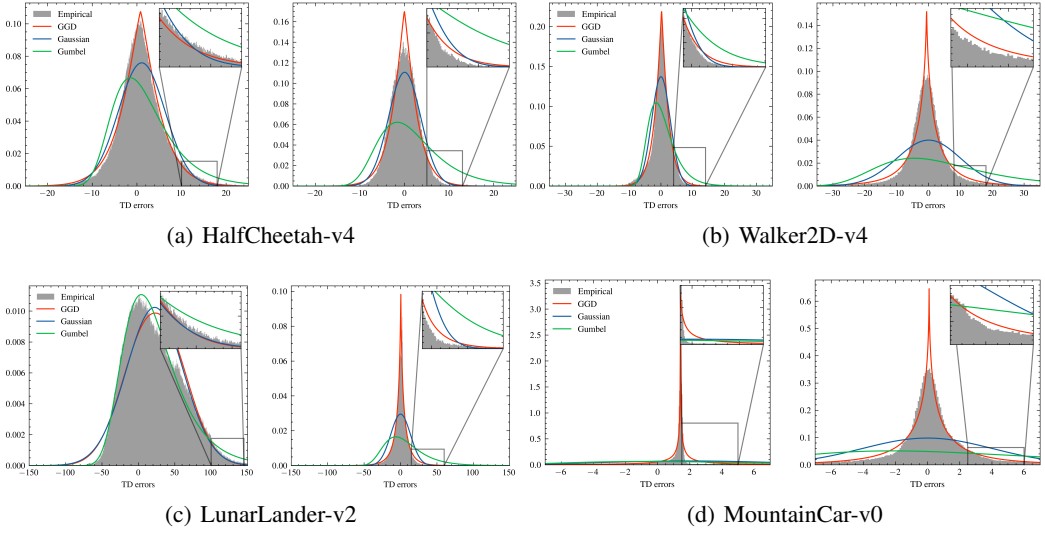

Figure 7:  TD error plots of PPO.

## A.2 COEFFICIENTS OF VARIATION OF PARAMETER ESTIMATION

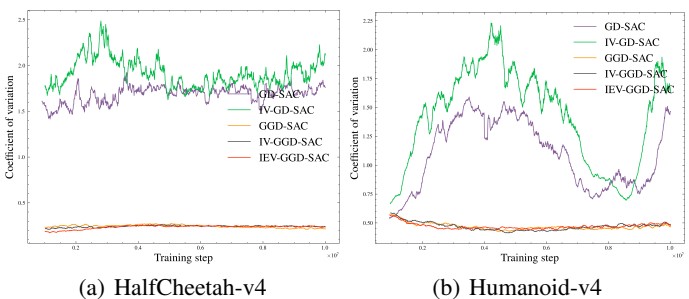

(a) HalfCheetah-v4          (b) Humanoid-v4

Figure 8: Coefficients of variation of parameter estimates for SAC variants.

## A.3 PPO ON OTHER ENVIRONMENTS

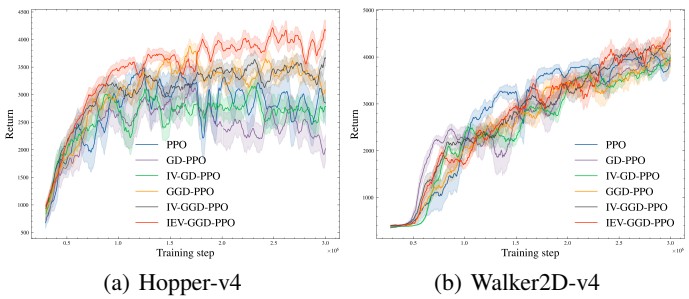

(a) Hopper-v4          (b) Walker2D-v4

Figure 9: Sample efficiency curves of PPO on remained environments.

## A.4 PARAMETER ESTIMATION

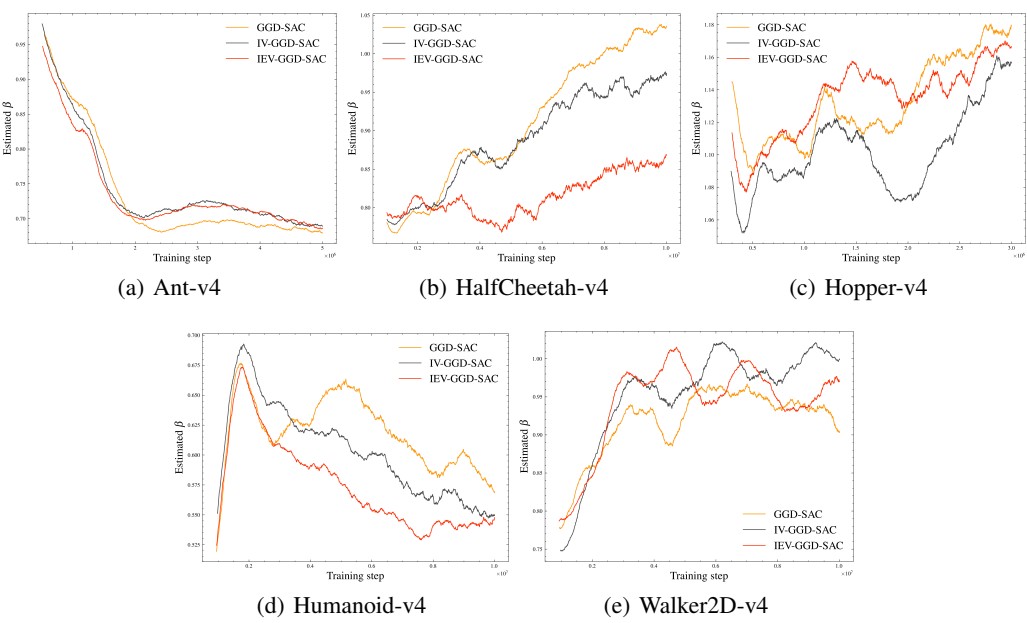

(a) Ant-v4          (b) HalfCheetah-v4          (c) Hopper-v4

(d) Humanoid-v4          (e) Walker2D-v4

Figure 10: Estimated $\beta$ for SAC variants.

The parameter estimates from the beta head reveal consistently low values of $\beta$ across all environments, indicating a leptokurtic TD error distribution, as depicted in Figure 10. These findings align with the observations from the TD error plots, where TD errors exhibit a closer resemblance to the GGD rather than a Gaussian distribution.

# B PROOFS

## B.1 OF PROPOSITION 1

*Proof.* Consider a finite sample $X_1, X_2, ..., X_n$ of independent, normally distributed random variables with $X \sim \mathbb{P}_{\theta_0}$, where $\theta_0 = (\mu, \sigma) \in \Theta$ represents the true generative parameters. Under this assumption, both skewness $\gamma$ and kurtosis $\kappa$ are zero. Consequently, the moments are given by $\mathbb{E}[X] = \mu$, $\mathbb{E}\left[(X - \mu)^2\right] = \sigma^2$, $\mathbb{E}\left[(X - \mu)^3\right] = \sigma^{3/2}\gamma = 0$ and $\mathbb{E}\left[(X - \mu)^4\right] = \sigma^4(\kappa + 3) = 3\sigma^4$.

The MLE estimator of $\mu$ and $\sigma$ is $\hat{\theta} = (\hat{\mu}, \hat{\sigma})$, given by

$$\hat{\mu} = \bar{X} = \frac{1}{n}\sum_{i=1}^{n} X_i, \quad \text{and} \quad \hat{\sigma}^2 = \frac{1}{n}\sum_{i=1}^{n}(X_i - \bar{X})^2.$$

It is well known to be consistent (Casella & Berger, 2002).

Assuming $\hat{\theta} \xrightarrow{p} \theta_0$, where $\xrightarrow{p}$ denotes the convergence in probability, under appropriate regularity conditions, then the asymptotic normality theorem of Cramer leads to

$$\sqrt{n}\left(\hat{\theta} - \theta_0\right) \xrightarrow{p} \mathcal{N}(0, \mathcal{I}(\theta_0)^{-1}),$$

as $n \to \infty$ (Ferguson, 2017). Here, $\mathcal{I}(\theta_0) = \begin{pmatrix} \mathcal{I}_{11} & \mathcal{I}_{12} \\ \mathcal{I}_{21} & \mathcal{I}_{22} \end{pmatrix}$ is the Fisher information matrix.

Through straightforward calculations involving the log likelihood function derivatives, we obtain

$$\mathcal{I}(\theta_0) = \begin{pmatrix} \mathcal{I}_{11} & \mathcal{I}_{12} \\ \mathcal{I}_{21} & \mathcal{I}_{22} \end{pmatrix} = \begin{pmatrix} 1/\sigma^2 & 0 \\ 0 & 1/2\sigma^4 \end{pmatrix}.$$

Thus, as $n \to \infty$, $\sqrt{n}(\hat{\theta} - \theta_0) \xrightarrow{p} \mathcal{N}\left(0, \begin{pmatrix} \sigma^2 & 0 \\ 0 & 2\sigma^4 \end{pmatrix}\right)$.

Now, consider a finite sample $X_1, X_2, ..., X_n$ of independent, non-normally distributed random variables, with MLE estimator $\hat{\theta} = (\hat{\mu}, \hat{\sigma})$. By applying analogous reasoning, we seek the asymptotic distribution of $\sqrt{n}(\hat{\theta} - \theta_0)$. Letting $\tilde{\sigma}^2 = \frac{1}{n}\sum_{i=1}^{n}(X_i - \mu)^2$ gives us

$$\hat{\sigma}^2 = \frac{1}{n}\sum_{i=1}^{n}(X_i - \hat{\mu})^2$$

$$= \frac{1}{n}\sum_{i=1}^{n}(X_i - \mu)^2 - (\hat{\mu} - \mu)^2$$

$$= \tilde{\sigma}^2 - \frac{1}{n}\sum_{i=1}^{n}(\hat{\mu} - \mu)^2.$$

Therefore, as $\sqrt{n}(\hat{\mu} - \mu)^2 \xrightarrow{p} 0$,

$$\sqrt{n}(\hat{\theta} - \theta_0) = \sqrt{n}\left(\hat{\mu} - \mu, \hat{\sigma}^2 - \sigma^2\right)$$
$$= \sqrt{n}\left(\hat{\mu} - \mu, \tilde{\sigma}^2 - \sigma^2\right) - \left(0, \sqrt{n}(\hat{\mu} - \mu)^2\right) \quad (11)$$
$$\xrightarrow{p} \sqrt{n}\left(\hat{\mu} - \mu, \tilde{\sigma}^2 - \sigma^2\right).$$

Denoting $\tilde{\theta} = (\hat{\mu}, \tilde{\sigma}^2)$, Equation (11) states that $\hat{\theta}$ and $\tilde{\theta}$ are asymptotically equivalent, i.e.,

$$\left(\hat{\theta} - \tilde{\theta}\right) \xrightarrow{p} 0.$$

By Slutsky's theorem (Ferguson, 2017), $\tilde{\theta}$ has the same asymptotic normality as $\hat{\theta}$, i.e.,

$$\sqrt{n}\left(\tilde{\theta} - \theta_0\right) \xrightarrow{p} \mathcal{N}(0, \mathcal{K}),$$

where $\mathcal{K} = \begin{pmatrix} \mathcal{K}_{11} & \mathcal{K}_{12} \\ \mathcal{K}_{21} & \mathcal{K}_{22} \end{pmatrix}$ is a covariance matrix with

$$\begin{aligned}
\mathcal{K}_{11} &= \mathbb{E}\left[(X_i - \mu)^2\right] = \sigma^2, \\
\mathcal{K}_{12} &= \mathbb{E}\left[(X_i - \mu)((X_i - \mu)^2 - \sigma^2)\right] = \mathbb{E}\left[(X_i - \mu)^3\right] = \sigma^{\frac{3}{2}}\gamma, \\
\mathcal{K}_{21} &= \mathcal{K}_{12}, \text{ and} \\
\mathcal{K}_{22} &= \mathbb{E}\left[((X_i - \mu)^2 - \sigma^2)((X_i - \mu)^2 - \sigma^2)\right] \\
&= \mathbb{E}\left[(X_i - \mu)^4\right] - \sigma^4 = \sigma^4(\kappa + 2).
\end{aligned}$$

From the equality between $\mathcal{K}$ and $\mathcal{I}(\theta_0)^{-1}$, we find that

$$\sigma^4(\kappa + 2) = 2\sigma^4 \iff \kappa = 0.$$

Hence, when normality is assumed for non-normally distributed data, the bias of the standard error estimate $\hat{\sigma}^2$ based on $\mathcal{I}^{-1}$ depends on $\kappa$, i.e., negative when $\kappa > 0$ and positive when $\kappa < 0$. $\qquad\square$

### B.2 OF THEOREM 1

To prove that the NLL function of GGD is well defined for $\beta \in (0, 2]$, we show that the PDF of GGD is everywhere positive with it being a positive definite function. An easy but effective proof can be done by demonstrating that the PDF of GGD for $\beta \in (0, 2]$ is equivalent to the characteristic function of an $\alpha$-stable distribution, up to a normalizing constant (Dytso et al., 2018). Given the positive definiteness of all characteristic functions (Ushakov, 2011; Dytso et al., 2018), this equivalence assures the positive definiteness of the GGD PDF. However, we offer a proof rooted in the properties of the positive definite function class.

*Proof.* To demonstrate the positive definiteness of the GGD PDF, it is sufficient to show the positivity of the function:

$$f_\beta(x) = e^{-|x|^\beta},$$

where $\beta \in (0, 2]$.

For a class of positive definite functions $\mathcal{F}$, which can be interpreted as Fourier transforms of bounded non-negative distributions, the function $f \in \mathcal{F}$, i.e.,

$$f(x) = \int_\infty^\infty e^{i\chi x} dV(\chi),$$

satisfy the following properties:

1. For any non-negative scalars $a_1, a_2$ and functions $f_1, f_2 \in \mathcal{F}$, $a_1 f_1 + a_2 f_2 \in \mathcal{F}$.

2. For $f_1, f_2 \in \mathcal{F}$, $f_1 f_2 \in \mathcal{F}$.

3. If a sequence of functions $f_n \in \mathcal{F}$ converges uniformly in every finite interval, then the limit function $\lim_{n\to\infty} f_n \in \mathcal{F}$.

Now, let $\rho = \beta/2$, and we aim to prove that $f_\rho = \exp(-|x|^{2\rho})$ belongs to $\mathcal{F}$ for $\rho \in (0, 1)$, excluding the trivial case $\rho = 1$. Since for $0 < \rho < 1$,

$$|x|^{2\rho} = c_\rho \int_0^\infty \frac{\chi^{2\rho-1} d\chi}{1 + (\frac{\chi}{x})^2}, \quad c_\rho > 0,$$

$f_\rho$ can be expressed as a uniform limit of functions $f_\rho \approx \lim_{n\to\infty} f_n$, where each $f_n$ takes the form:

$$
\begin{aligned}
f_n &= \exp\left(-\sum_{\nu=1}^{n} \frac{a_\nu^2}{1 + \left(\frac{b_\nu}{x}\right)^2}\right) \\
&= \exp\left(-\sum_{\nu=1}^{n} \frac{a_\nu^2 x^2}{x^2 + b_\nu^2}\right) \\
&= \prod_{\nu=1}^{n} \exp\left(-\frac{a_\nu^2 x^2}{x^2 + b_\nu^2}\right),
\end{aligned}
$$

for some sequences $\{a_\nu\}$ and $\{b_\nu\}$ with $\nu \in \{1, ..., n\}$.

By simplifying, we find that for some $a \in \{a_\nu\}$ and $b \in \{b_\nu\}$ and letting $c^2 = a^2 b^2$:

$$
\begin{aligned}
\exp\left(-\frac{a^2 x^2}{x^2 + b^2}\right) &= \exp\left(\frac{-a^2 x^2 - a^2 b^2 + a^2 b^2}{x^2 + b^2}\right) \\
&= \exp\left(-\frac{a^2(x^2 + b^2)}{x^2 + b^2} + \frac{a^2 b^2}{x^2 + b^2}\right) \\
&= \exp\left(-a^2\right)\exp\left(\frac{c^2}{x^2 + b^2}\right).
\end{aligned}
$$

Note that $\exp\left(c^2/(x^2 + b^2)\right) = \sum_{n=0}^{\infty} c^{2n}/n! \times (x^2 + b^2)^{-n}$ from Taylor expansions.

From the properties of a positive definite function class, it is now sufficient to show that $(x^2 + b^2)^{-1}$ belongs to $\mathcal{F}$ to prove $f_\beta \in \mathcal{F}$. We demonstrate this by expressing it as a Fourier transform of a bounded non-negative distribution:

$$
(x^2 + b^2)^{-1} = \frac{1}{2b} \int_{-\infty}^{\infty} e^{i\chi x} e^{-b|\chi|} d\chi.
$$

$\square$

### B.3 OF THEOREM 2

*Proof.* For random variable $X \sim \mathrm{GGD}(0, \alpha, \beta)$, its cumulative distribution function $F_X(t)$ is defined as

$$
F_X(t) = \frac{1}{2} + \mathrm{sign}(t)\frac{\gamma\left(1/\beta, (|t - \mu|/\alpha)^\beta\right)}{2\Gamma(1/\beta)},
$$

where

$$
\Gamma(x) = \int_0^\infty t^{x-1} e^{-t} dt, \quad \text{and} \quad \gamma(x, s) = \int_0^s t^{x-1} e^{-t} dt,
$$

represent the gamma function and lower incomplete gamma function for a complex parameter $x$ with a positive real part.

Expanding the left-hand side of Equation (7), we obtain

$$
\begin{aligned}
\int_{-\infty}^{x} F_{X_1}(t) - F_{X_2}(t)dt &= \int_{-\infty}^{x} \mathrm{sign}(t)\left(\frac{\gamma\left(1/\beta_1, (|t - \mu|/\alpha)_1^\beta\right)}{2\Gamma(1/\beta_1)} - \frac{\gamma\left(1/\beta_2, (|t - \mu|/\alpha)_2^\beta\right)}{2\Gamma(1/\beta_2)}\right) dt \\
&= \int_{x}^{\infty} \frac{\gamma\left(1/\beta_2, (|t - \mu|/\alpha)_2^\beta\right)}{2\Gamma(1/\beta_2)} - \frac{\gamma\left(1/\beta_1, (|t - \mu|/\alpha)_1^\beta\right)}{2\Gamma(1/\beta_1)} dt.
\end{aligned}
$$

$$(12)$$

Defining $f(\beta, t) = \gamma(1/\beta, (|t - \mu|/\alpha)^\beta)/2\Gamma(1/\beta)$, we aim to demonstrate the monotonicity of $f$ to conclude the proof, as monotonically increasing $f$ leads to $f(\beta_2, x) - f(\beta_1, x) \geq 0$ for $\beta_2 \geq \beta_1$, i.e., the integral in Equation (12) is greater than or equal to zero.

With the definition of the gamma and lower incomplete gamma function,

$$f(\beta, t) = \frac{\gamma\left(1/\beta, (|t - \mu|/\alpha)^\beta\right)}{2\Gamma(1/\beta)} = \frac{\int_0^{(x/\alpha)^\beta} t^{1/\beta - 1} e^{-t} dt}{\int_0^\infty t^{1/\beta - 1} e^{-t} dt}. \tag{13}$$

Employing integration by substitution with $u = \left(\alpha^\beta t\right)^{\frac{1}{\beta}}$, Equation (13) transforms to

$$f(\beta, t) = \frac{\int_0^x \beta/\alpha \left((u/\alpha)^\beta\right)^{1/\beta - 1} e^{-(u/\alpha)^\beta} \left(\left((u/\alpha)^\beta\right)^{1/\beta - 1}\right)^{-1} du}{\int_0^\infty \beta/\alpha \left((u/\alpha)^\beta\right)^{1/\beta - 1} e^{-(u/\alpha)^\beta} \left(\left((u/\alpha)^\beta\right)^{1/\beta - 1}\right)^{-1} du}$$

$$= \frac{\int_0^x e^{-(u/\alpha)^\beta} du}{\int_0^\infty e^{-(u/\alpha)^\beta} du}.$$

The function $f$ is thus increasing if, for $\beta_1 \leq \beta_2$ and any $\alpha > 0$,

$$\frac{\int_0^x e^{-(u/\alpha)^{\beta_1}} du}{\int_0^\infty e^{-(u/\alpha)^{\beta_1}} du} \leq \frac{\int_0^x e^{-(u/\alpha)^{\beta_2}} du}{\int_0^\infty e^{-(u/\alpha)^{\beta_2}} du}$$

$$\iff \int_0^x \int_0^\infty e^{-\left((u/\alpha)^{\beta_1} + (v/\alpha)^{\beta_2}\right)} dv du \leq \int_0^x \int_0^\infty e^{-\left((u/\alpha)^{\beta_1} + (v/\alpha)^{\beta_2}\right)} du dv,$$

which follows from the monotonicity of the exponential function (Dytso et al., 2018).

Consequently, $f(\beta_2, t) - f(\beta_1, t) \geq 0$, implying that Equation (12) is greater than or equal to zero. $\qquad \square$

### B.4 OF PROPOSITION 2

*Proof.* It is well known that the variance of the unbiased estimator $s^2$ is given by

$$\mathbb{V}[s^2] = \frac{1}{n}\left[\kappa - \frac{n-3}{n-1}\sigma^2\right].$$

And the MSE of a biased estimator $s_\omega^2 = \omega(n-1)s^2$ is

$$\mathrm{MSE}(s_\omega^2) = \omega^2(n-1)^2 \mathbb{V}[s^2] + \left[(n-1)\omega - 1\right]^2 \sigma^4. \tag{14}$$

By differentiating Equation (14) with respect to $\omega$, we can calculate the optimal $\omega$ value with minimal $\mathrm{MSE}(s_\omega^2)$ as

$$\left.\frac{d\,\mathrm{MSE}(s_\omega^2)}{d\omega}\right|_{\omega = \omega^*} = 2(n-1)^2\omega^*\mathbb{V}[s^2] + 2\left[(n-1)\omega^* - 1\right]\sigma^4 = 0.$$

Therefore,

$$\omega^* = \left[\frac{\sigma^4}{(n-1)\left(\mathbb{V}[s^2] + \sigma^4\right)}\right] = \left[\frac{n-1}{n}\kappa + (n+1)\right]^{-1}.$$

It is easy to show that this is the optimal value, given that the second derivative is positive, i.e.,

$$\frac{d^2\,\mathrm{MSE}(s_\omega^2)}{d\omega^2} = 2(n-1)^2\mathbb{V}[s^2] + 2(n-1)^2\sigma^4 > 0.$$

$\qquad \square$

## C EXPERIMENTAL DETAILS

All experiments are conducted on a computational infrastructure consisting of 8 NVIDIA A100 80GB PCIe GPUs and 256 AMD EPYC™ 7742 processors. Further details on the software setup will be made available openly through GitHub following the completion of the peer-review process.

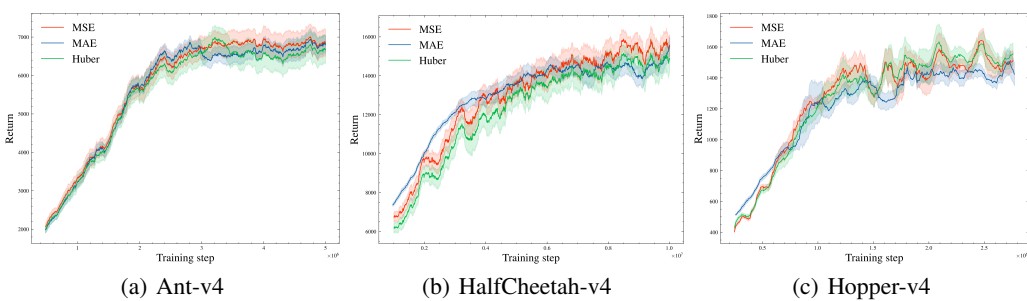

| (a) Ant-v4 | (b) HalfCheetah-v4 | (c) Hopper-v4 |

Figure 11: Ablation study on loss selection for BIEV regularization.

**Additional noise** To extend our study to discrete control scenarios, we enhance the baseline environments with stochastic perturbations. In CartPole-v1 and MountainCar-v0, we introduce uniform noise to manipulate forces or torques. Furthermore, in LunarLander-v2, wind dynamics are activated. For a comprehensive analysis of how wind impacts the LunarLander-v2 dynamics, please consult the official documentation[1].

**Implementation** To bolster numerical stability and enforce positivity constraints, we apply the softplus function, a smooth approximation to the ReLU function (Dugas et al., 2000), to the outputs from the variance or beta head. Furthermore, we modify the NLL loss for the GGD by employing $Q^\beta$ as a multiplier rather than as an exponent:

$$\mathcal{L}_{\text{GGD-NLL}} \approx \sum_t \left( |\delta_t| / Q_t^\alpha \right) \times Q_t^\beta - \log Q_t^\beta / Q_t^\alpha + \log \Gamma(1/Q_t^\beta),$$

Although this adaptation deviates from the exact formulation of the NLL loss of the GGD, the difference in computed loss between the original and modified forms remains negligible across practical ranges of the TD error and $\beta$. This adjustment offers a practical advantage by addressing the issue of flat regions in the original loss function, thus yielding more informative gradients for model updates. Importantly, despite this modification, the positive-definiteness ensured by Theorem 1 is preserved, as the modified loss remains proportionally related to the original when $\beta$ is set to 1. This property is crucial for maintaining stability and convergence in error modeling based on GGD.

Additionally, introducing scale invariance into the loss framework can be achieved by a simple adjustment to the regularization term: replacing the squared TD error term with the absolute error. As the BIEV regularization framework is agnostic to the specific choice of loss function, the impact of the loss selection on the overall performance of BIEV regularization is minimal, as demonstrated in Figure 11. With MAE loss, the resulting loss function depends only on the absolute TD errors, reducing the influence of the regularization temperature $\lambda$. In contrast, for other loss functions, the performance tends to be more sensitive to the choice of temperature, motivating us to perform an ablation study on the effect of the regularization temperature, detailed in Appendix D.3.

# D ABLATION STUDIES

We conduct a series of ablation studies using the SAC algorithm across selected MuJoCo environments.

## D.1 ON RISK-AVERSE WEIGHTING

The theoretical foundation of the risk-averse weighting $\omega_t^{\text{RA}} = Q_t^\beta$ is provided by Theorem 2. Its empirical effectiveness, in comparison to the original GGD NLL and risk-seeking weighting $\omega_t^{\text{RA}} = 1/Q_t^\beta$, is demonstrated in Figure 12. It is evident that risk-aversion fosters sample-efficient training but does not necessarily translate to improved asymptotic performance. Notably, the adoption of

---

[1]https://gymnasium.farama.org/environments/box2d/lunar_lander

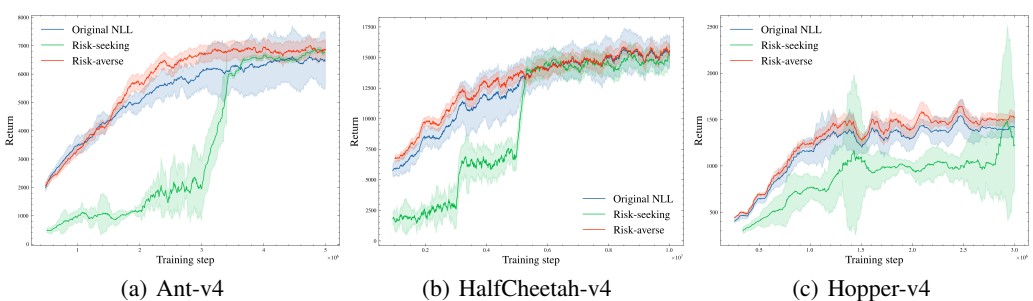

Figure 12: Ablation study on risk-averse weighting.

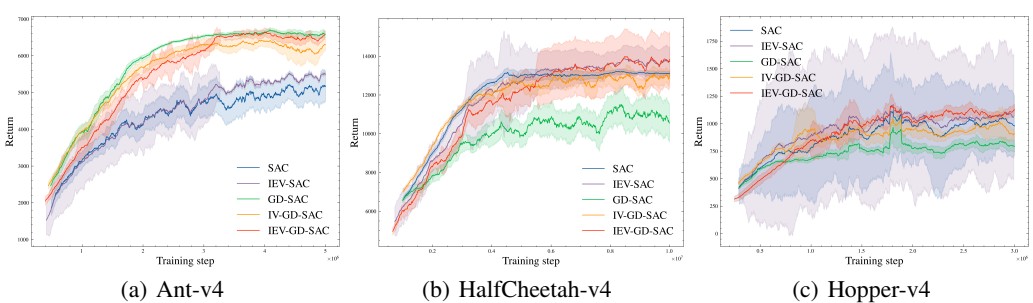

Figure 13: Ablation study on BIEV regularization.

risk-seeking behavior results in periodic performance jumps, indicating active exploration during side-stepping.

## D.2 ON BIEV REGULARIZATION

The BIEV loss term is inherently independent of the parameter head, as it relies solely on the empirical variance of TD errors. This encourages us to investigate its efficacy independently of the GGD error modeling scheme. As depicted in Figure 13, the impact of BIEV on sample efficiency is nearly equivalent to that of BIV regularization, but slightly more advantageous in terms of asymptotic performance.

## D.3 ON REGULARIZING TEMPERATURE

The loss function presented in Equation (4) is influenced by the scale of the rewards, requiring the regularizing temperature to be adjusted accordingly for different environments. As shown in Figure 14, the regularizing temperature $\lambda$ affects sample efficiency, training stability, and asymptotic performance. However, this sensitivity is notable only with large differences in scale, such as between 0.01 and 100. Furthermore, as the regularizing temperature approaches zero, the model's performance converges to that of the GGD error modeling scheme without BIEV regularization, labeled as '0', as expected.

## D.4 ON ALPHA HEAD

We exclusively utilize the beta head for computational efficiency and training stability, as elaborated in Remark 2. Integrating the alpha head, as anticipated, diminishes sample efficiency and even leads to decreased asymptotic performance, particularly evident in Ant-v4, as depicted in Figure 15.

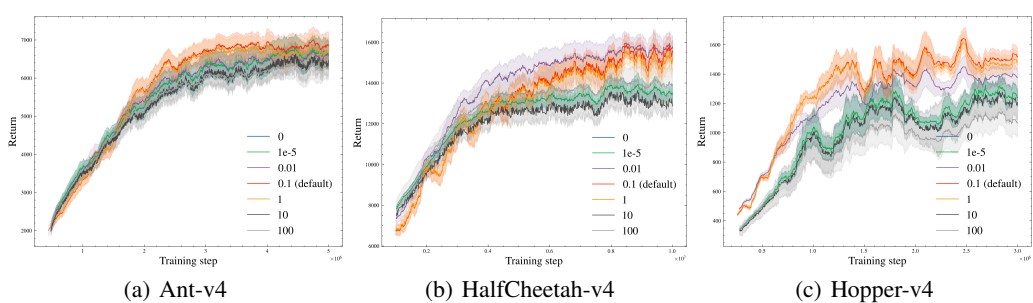

|              |                   |                |
|:------------:|:-----------------:|:--------------:|
| (a) Ant-v4   | (b) HalfCheetah-v4 | (c) Hopper-v4  |

Figure 14: Ablation study on regularizing temperature $\lambda$.

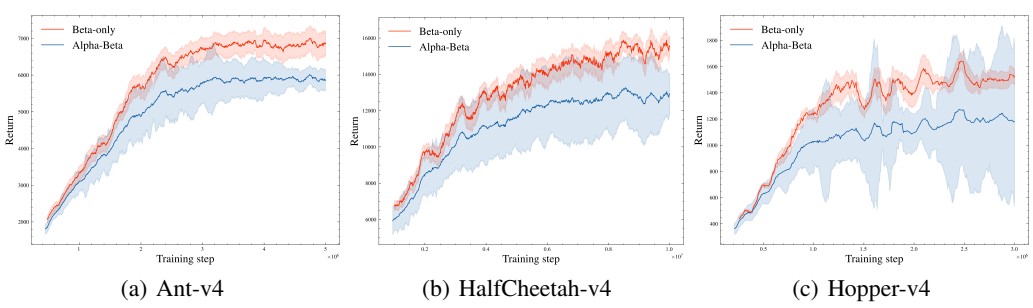

|              |                   |                |
|:------------:|:-----------------:|:--------------:|
| (a) Ant-v4   | (b) HalfCheetah-v4 | (c) Hopper-v4  |

Figure 15: Ablation study on alpha head.

## E EXTENSION

As discussed in Section 5, we expand our empirical analysis to include $Q$-learning, with particular attention to DQN (Mnih et al., 2015). Figure 16 demonstrates that TD errors generated by DQN exhibit deviations from a Gaussian distribution and closely follow the GGD.

The performance of DQN in noisy discrete control environments, as depicted in Figure 17, highlights that simply applying our method to $Q$-learning is not sufficiently effective. Interestingly, regularization appears to have minimal impact, as evidenced by the similar performance observed among DQN variants equipped with GGD error modeling.

Since our implementation of DQN utilizes a greedy action policy, the approximated value function inherently represents the policy. Consequently, employing risk-averse weighting might impede exploration. Conversely, in policy gradient algorithms featuring separate policy networks, such risk-aversion serves to mitigate noisy supervision, aligning with its intended purpose. The adverse effects of risk-aversion are evident in Figure 18, where the model trained with the original NLL exhibits slightly superior sample efficiency compared to the proposed method.

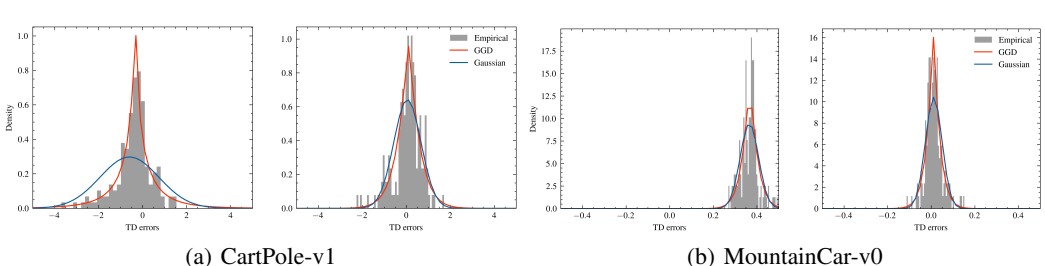

(a) CartPole-v1

(b) MountainCar-v0

Figure 16: TD error plots of DQN.

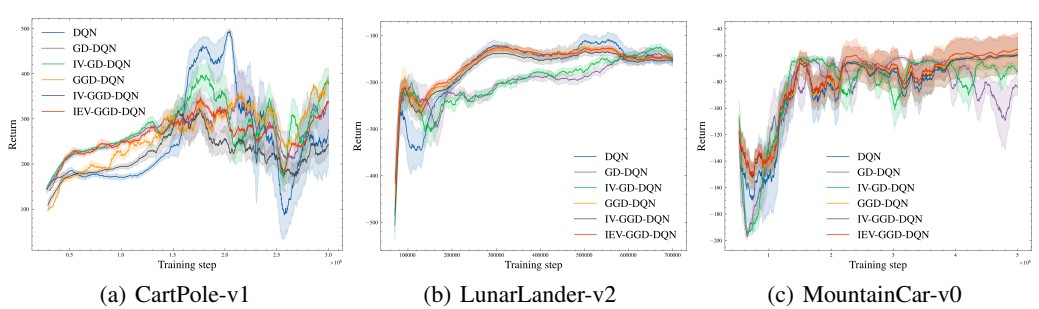

(a) CartPole-v1

(b) LunarLander-v2

(c) MountainCar-v0

Figure 17: Sample efficiency curves of DQN on noisy discrete control environments.

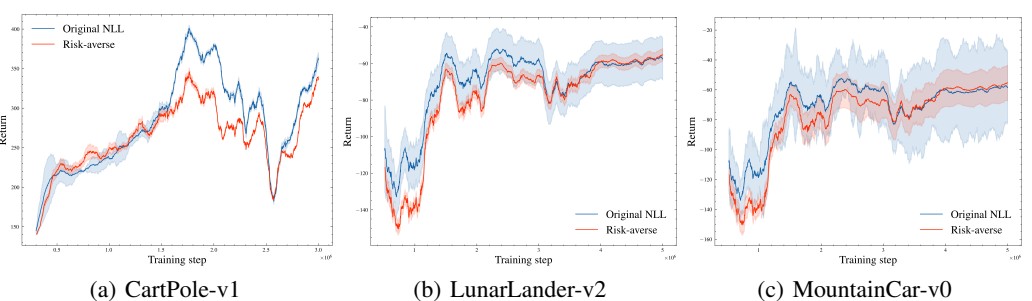

(a) CartPole-v1

(b) LunarLander-v2

(c) MountainCar-v0

Figure 18: Ablation study on risk-averse weighting for DQN.

