# OpenReview forum: "Generalized Gaussian Temporal Difference Error for Uncertainty-aware Reinforcement Learning"
_ICLR.cc/2025/Conference — Submitted to ICLR 2025_

### Official Review · Reviewer_BUWP · 2024-11-02

**Soundness:** 3
**Presentation:** 1
**Contribution:** 2
**Rating:** 3
**Confidence:** 4

**Summary:**

Instead of Gaussian distribution, this paper applies generalized Gaussian distribution (GGD) to model temporal difference (TD) error, and proposes an uncertainty-aware objective function to minimize the TD error. It discussed several properties of the GGD distributions and empirically the proposed objective can work better than some baselines on common benchmark RL problems.

**Strengths:**

- Discussion on some advantageous properties of GGD modelling
- Empirical success with the proposed objective

**Weaknesses:**

The main problem with the current paper is its presentation, especially the lack of motivation for the objective function.

1. Fig.2 requires further clarification. It is surprising to see that, for example, the Gaussian fitted PDF is so different from the empirical histogram for Ant-v4 (see also the MountainCar-v0 in Fig.7(d) for a more obvious mismatch). For Hopper-v4, the standard deviations of the two Gaussians are very close to each other despite the empirical histograms having very different shapes. More importantly, this figure is about the distribution of ALL TD errors across state-action pairs, while the NLL objective is about the underlying distribution of ONE single state-action pair. For this figure to make sense, one needs to assume that all TD errors come from the same underlying distribution, which doesn’t seem to be the case as Eq.(4) uses different betas for different step t.

2. L273 offers a hypothesis, which is unverified. How do we define “unexpected states and rewards” and could you provide empirical support for this? Similarly, the explanation of interplay is unsatisfactory for the paragraph in L278. No evidence to support claims like “As training advances, aleatoric errors tend to become more influential while epistemic uncertainty diminishes, potentially resulting in non-normally distributed errors with heavier tails.”

3. It remains unclear how Thm.2 can lead to the specific weighting in Eq.(4). A detailed derivation would be helpful. Similarly, it is unclear why Eq.(8) can address the bias issue mentioned in L332. The current paper has too many hand-wavy arguments without sufficient rigorous discussions.

4. Finally, it seems that all theoretical properties in the current paper are stemmed from prior work, as shown by the reference in every theorem/proposition. If this is indeed the case, then what would be the main theoretical contribution of the current paper, other than an application of an existing tool (GGD) to TD learning?

**Questions:**

1. Please explain further why the Gaussian fits in Fig.2 are very off, and why looking at the TD errors of all examples can justify the underlying GGD assumption as discussed above.

2. How can Thm.2 lead to the specific weighting in (4)?

3. What is the main theoretical contribution of this work?

---

> ### Author Response · Authors · 2024-11-25
>
> ## On the Theoretical Contributions
>
> > Related to Weakness 4 and Question 3
>
> We appreciate the reviewer's insights and acknowledge the need to delineate our theoretical contributions more clearly.
> While our work builds upon existing mathematical tools like the generalized Gaussian distribution (GGD) and uncertainty modeling in temporal difference (TD) learning, it introduces novel contributions specific to reinforcement learning (RL):
>
> - **Empirical Demonstration of Limitation in Prior Models**:
>   Section 3.1.1 empirically shows significant deviations from Gaussian assumptions of TD error distributions across various RL tasks, highlighting the necessity for more flexible modeling.
>   This foundational observation was not established or systematically analyzed in prior works.
> - **Novel Application of GGD for Uncertainty Mitigation**:
>   We extend the GGD to aleatoric uncertainty modeling, providing a closed-form relationship between the shape parameter $\beta$ and uncertainty (Section 3.1).
>   This integration, tailored for heteroscedastic TD errors, advances beyond traditional variance-based approaches.
> - **Theoretically Grounded Weighting Scheme**:
>   As detailed in Section 3.2, we propose a new batch inverse error variance (BIEV) weighting mechanism that incorporates kurtosis considerations for epistemic uncertainty.
>   This theoretical enhancements refines existing weighting strategies by explicitly accounting for tail behavior.
> - **Risk-Averse Weighting via Stochastic Dominance**:
>   Leveraging second-order stochastic dominance properties of GGD (Theorem 2), we develop a risk-averse weighting strategy.
>   This is a novel integration of SSD principles into RL for robust performance under heteroscedastic conditions.
> - **Broader Insights for Robust and Risk-Sensitive RL**:
>   Our insights provide theoretical and practical tools for designing robust RL algorithms that effectively handle tailed error distributions and improve epistemic uncertainty estimation (Section 5).
>
> In summary, while some theoretical tools stems from prior work, their specific integration into TD learning and reinforcement learning tasks, along with the proposed theoretical and practical innovations, constitute the primary contributions of this work.
> These developments improve performance significantly and address critical limitations in existing uncertainty-aware RL methods, as evidenced by our experimental results in Section 4.

---

> ### Author Response · Authors · 2024-11-25
>
> ## On Uncertainty Dynamics
>
> > Related to Weaknesses 2 and 3
>
> Thank you for the opportunity to clarify.
>
> We define "unexpected states and rewards" as regions encountered during exploration that deviate from states and rewards sampled by the agent's policy during convergence.
> Empirical support for this hypothesis is presented in Figure 2, which illustrates the evolution of TD error distributions during training.
> Specifically, in earier phases, broader and heavier-tailed distributions reflect the agent's exploration of less familiar states and rewards.
> Figures 6 and 7 also support these trends across different environments.
>
> We agree that the explanation in Line 278 could benefit from additional evidence.
> The claim regarding aleatoric and epistemic uncertainty dynamics is supported by:
>
> 1. **Theoretical Support**:
>    As par [Kendall & Gal, 2017](https://arxiv.org/abs/1703.04977), epistemic uncertainty typically diminishes as the agent collects more data, narrowing the parameter space for value function approximation.
>    On the other hand, aleatoric uncertainty, arising from environmental stochasticity, remains irreducible and becomes more prominent as training progresses.
> 2. **Empirical Evidence**:
>    Figures 4 and 10 provide evidence for this interplay.
>    The convergence of $\beta$ estimates toward lower and leptokurtic values (Figure 10) and the reduced coefficient of variation for $\beta$ (Figure 4) indicate stabilized estimates of tail behavior as epistemic uncertainty decreases during training.
>    Specifically, Figure 4 illustrates the coefficients of variation (CV) for parameter estimates of $\beta$ during training.
>    CV is defined as the ratio of the standard deviation to the mean of parameter estimates, offering a scale-invariant measure of stability.
>    The CV values for $\beta$ consistently decrease as training progresses across all environments, reflecting greater stability in the estimation of the shape parameter.
>    Epistemic uncertainty, which arises from a lack of sufficient data or exploration, manifests as high variance in parameter estimates during early training phases.
>    As training progresses and the agent collects more data, this uncertainty diminishes, leading to more stable and consistent estimates of $\beta$, as shown by the reduced CV.
>    On the other hand, Figure 6 and 7 tracks the evolution of TDE over training epochs for SAC and PPO variants.
>    Across all environments, the experiments demonstrate that the TD error distributions become more leptokurtic (heavy-tailed) as training progresses, with decreasing $\beta$ value.
>    Additionally, Figure 10 and our additional experiments suggests that in most cases, $\beta$ estimates decrease and converge within the leptokurtic bound.
>    This trend indicates that the TD error distributions become more leptokurtic (heavy-tailed) as training progresses.
>    The closed form of aleatoric uncertainty suggest that aleatoric uncertainty exhibits a negative proportionality to the shape parameter $\beta$ on an exponential scale, thus aligning with the hypothesis that aleatoric uncertainty increasingly dominates the error structure.
>
> We revised the relevant paragraphs accordingly.
>
> Regarding Equation (8), our BIEV mechanism addresses bias by assuming constant approximation bias in TD errors, as suggested in [Flennerhag et al., 2020](https://arxiv.org/abs/2010.02255).
> This assumption allows us to focus on variance estimation to minimize the impact of bias on the overall uncertainty-aware objective.

---

> ### Author Response · Authors · 2024-11-25
>
> ## On Risk-averse Weighting
>
> > Related to Weakness 3 and Question 2
>
> Theorem 2 establishes second-order stochastic dominance (SSD) among GGD random variables, where distributions with larger $\beta$ values are preferable under a risk-averse framework due to their more less spread-out and predictable nature.
> The connection to Equation (4), $\omega^\text{RA}_t=Q^\beta_t$, is as follows:
>
> 1. Theorem 2 states that for two GGD variables $X_1\sim\text{GGD}(0,\alpha,\beta_1)$ and $X_2\sim\text{GGD}(0,\alpha,\beta_2)$ with $\beta_1 \leq \beta_2$, $X_2$ exhibits SSD over $X_1$.
>    Intuitively, this means that larger $\beta$ values lead to tighter, less dispersed distributions.
> 2. Considering the objective in risk-averse weighting, we prioritize more predictable samples (less dispersed, smaller aleatoric uncertainty) by assigning weights proportional to $\beta$.
>    A direct application of SSD implies that higher $\beta$ values should correspond to higher weights, encouraging the model to focus on less noisy samples.
> 3. Now formulating the weighting according to above, to balance these considerations in Equation (4), we define $\omega^\text{RA}_t=Q^\beta_t$, where $Q^\beta_t$ represents the local $\beta$ value for the TD error at step t.
>    This weighting ensures that samples with higher $\beta$ (less spread-out distributions) are weighted more heavily, aligning with the risk-averse preference established by Theorem 2.
>    In addition, by normalizing the weight of each sample in a batch relatively, weighting ensures that the model appropriately accounts for the reliability of the estimate from each data point, resulting in robustness against inaccuracies in $Q^\beta_t$ estimate.
>
> This weighting scheme aligns with SSD principles, leveraging GGD properties to construct a risk-averse weighting strategy that prioritizes less noisy samples.
>
> ## On Temporal Difference Error Distributions
>
> > Related to Weakness 1 and Question 1
>
> The mismatch of the Gaussian fitted PDF and the similar fitted variance in Hopper-v4 are due to the Gaussian distribution's inability to capture the higher-order moments of the empirical histograms.
> These observations underscore one of the paper's key motivations, the inadequacy of assuming normally distributed TD errors, and necessitate a more flexible modeling framework, such as GGD-based error modeling.
> The GGD explicitly incorporates a shape parameter $\beta$ that captures kurtosis and tail behavior, enabling it to model distributions like those in Hopper-v4, where higher-order moments play a critical role in distinguishing between training phases.
>
> We agree that the NLL objective optimizes the distribution of TD errors for individual state-action pairs, not the global distribution.
> However, Figure 2 is intended to provide approximates of the overall trend, showing that even when aggregated, the TD errors exhibit significant deviation from normality.
> This aggregated view is a useful diagnostic for highlighting general trends, such as non-Gaussian characteristics and changes in tailedness over time, which motivate the need for a more expressive distributional model at the local level.
> Although such a plot cannot fully reflect the variability of individual distributions at each time step $t$, it effectively showcases the widespread non-normality of TD errors.
> Please note that we leveraged the property that the sum of GGD samples also follow GGD, despite the fact that aggregated generalized Gaussian distributed does not necessarily imply the generalized Gaussian property of each sample, to further supporting the aggregation of TD errors in Figure 2 to highlight the relevance of GGD modeling.

---

### Official Review · Reviewer_W9ns · 2024-11-02

**Soundness:** 3
**Presentation:** 1
**Contribution:** 2
**Rating:** 8
**Confidence:** 3

**Summary:**

In this work, the authors present a method to estimate uncertainty based on the Generalized Gaussian Distribution (GGD) to better characterize temporal difference error (TD-error) distributions. Unlike previous work, which assumes Gaussianity, the GGD captures additional parameters, specifically kurtosis, to account for higher-order moments, thus enhancing the description of the TD-error distribution. The authors then modify an existing algorithm to operate under GGD assumptions, demonstrating improved performance across various settings. They also provide theoretical guarantees for well-behaved probability density function optimization under GGD assumptions and offer insights into why their proposed method is effective.

**Strengths:**

- (S1): The authors provide strong motivation for improving uncertainty-aware reinforcement learning (RL), it is an active area with relevance in applications like risk-averse decision-making, managing the exploration-exploitation trade-off, and enhancing sample efficiency.

- (S2): They conduct extensive experimentation, including ablation studies and parameter sweeps, to demonstrate that their proposed method outperforms the baseline (the original uncertainty-aware method).

- (S3): The proposed method is supported by theoretical validations and guarantees. Additionally, the proofs and mathematical analyses are well-documented in the appendix, making them relatively straightforward to follow.

**Weaknesses:**

- (W1): The writing of the paper could be significantly improved. Many concepts and terms are used in the main text without proper definitions, making it difficult to follow.

- (W2): Some important aspects of the proposed method are not well-developed and are instead left to references from the baseline method. While this approach is generally acceptable, in this case, understanding key elements like the inclusion of GGD and the consideration of uncertainty during training requires substantial familiarity with the referenced material. As a result, the text is not self-contained, and combined with the clarity issues, makes the main content challenging to understand.

- (W3): Although extensive experimentation demonstrates the effectiveness of the proposal, some results are counterintuitive. There is also insufficient exposition of the proposed method’s inner workings and its differences from the baseline, for which performance plots alone are insufficient.

**Questions:**

- Q1: Related to W1 and W2, what is the intuition behind Equation 3? In the original paper (Mai et al., 2022), this is Equation (10), which is preceded by a detailed explanation of many complex terms in the equation. As it stands, I do not think the main text is self-contained, as it requires substantial prior knowledge of that specific reference.

- Q2: How are ensembles used in this method? Ensembles are mentioned, but there is no explanation of how they are applied. Is this related to the variance in epistemic uncertainty estimation in the regularization term?

- Q3: How does the regularization term capture epistemic uncertainty (as the uncertainty that can be reduced through learning)?

- Q4: In Equation 4, how are risk-averse weights introduced? Was this derived from the inclusion of GGD in Mai et al. (2022), or was it manually designed to incorporate risk sensitivity?

- Q5: Why is estimating only beta sufficient to account for uncertainty compared to variance? Since the GGD has three parameters, with the first two set to 0 and 1 respectively, how can variance and kurtosis be represented simultaneously by a single number, beta? If I understand correctly, variance and kurtosis can be derived from beta (and alpha, in the case of variance). How do both quantities vary for a fixed alpha and a changing beta? There may be a range of TD-error distributions that cannot be fully described by just beta, which could potentially decrease the agent's performance.

- Q6: If Equation 4 holds, why does optimizing for both alpha and beta jointly decrease the agent’s performance, as shown in Figure 15?

- Q7: If considering higher moments of the TD-error distribution improves performance, does this mean that greater improvements are seen in tasks where TD-error tails are larger or smaller? How does performance compare to the evolution of the moments in the observed TD-errors? Does this method show a greater improvement over the baseline when higher-order moments are more or less pronounced?

- Q8: Typo in lines 198 and 987, and the font size of labels and ticks in figures could be increased significantly to facilitate reading.

---

> ### Author Response · Authors · 2024-11-25
>
> ## Detailed Explanation of Equation 3
>
> > Relelated to Question 1
>
> We appreciate the reviewer highlighting the need for a more self-contained explanation of Equation (3).
> Among other parts, we acknowledge that Equation (3) may lack sufficient context, and we make the explanation more self-contained.
>
> ## On Epistemic Uncertainty Mitigation
>
> > Related to Questions 2 and 3
>
> We appreciate the opportunity to clarify the role of ensembles in our method and their connection to epistemic uncertainty estimation.
>
> Ensembles are employed in our framework to estimate epistemic uncertainty, which arises from limited exploration, i.e., insufficient data in the state-action space.
> Specifically, we use an ensemble of $k$ critics, each independently trained using the same TD error data but initialized with different random seeds.
> The variance among these critics' predictions $\mathbb{V}[\delta_t]$ serves as an empirical measure of epistemic uncertainty, capturing the disagreement across the ensemble.
> This ensemble-based variance is then used in the BIEV weighting in Equation (8).
> By penalizing high-variance predictions, the method prioritizes state-action pairs with more reliable, i.e., low-variance, estimates, improving robustness.
>
> ## On Risk-averse Weighting
>
> > Related to Weakness 3 and Questions 2 and 4
>
> Theorem 2 establishes second-order stochastic dominance (SSD) among GGD random variables, where distributions with larger $\beta$ values are preferable under a risk-averse framework due to their more less spread-out and predictable nature.
> The connection to Equation (4), $\omega^\text{RA}_t=Q^\beta_t$, is as follows:
>
> 1. Theorem 2 states that for two GGD variables $X_1\sim\text{GGD}(0,\alpha,\beta_1)$ and $X_2\sim\text{GGD}(0,\alpha,\beta_2)$ with $\beta_1 \leq \beta_2$, $X_2$ exhibit SSD over $X_1$.
>    Intuitively, this means that larger $\beta$ values lead to tighter, less dispersed distributions.
> 2. Considering the objective in risk-averse weighting, we prioritize more predictable samples (less dispersed, smaller aleatoric uncertainty) by assigning weights proportional to $\beta$.
>    A direct application of SSD implies that higher $\beta$ values should correspond to higher weights, encouraging the model to focus on less noisy samples.
> 3. Now formulating the weighting according to above, to balance these considerations in Equation (4), we define $\omega^\text{RA}_t=Q^\beta_t$, where $Q^\beta_t$ represents the local $\beta$ value for the TD error at step t.
>    This weighting ensures that samples with higher $\beta$ (less spread-out distributions) are weighted more heavily, aligning with the risk-averse preference established by Theorem 2.
>    In addition, by normalizing the weight of each sample in a batch relatively, weighting ensures that the model appropriately accounts for the reliability of the estimate from each data point, resulting in robustness against inaccuracies in $Q^\beta_t$ estimate.
>
> This weighting scheme aligns with SSD principles, leveraging GGD properties to construct a risk-averse weighting strategy that prioritizes less noisy samples.
>
> ## On Beta Estimation
>
> > Related to Question 5 and 6
>
> Thank you for this insightful question regarding the use of $\beta$ as the sole estimated parameter in our framework and trade-offs involved.
>
> While the GGD has three parameters, our method simplifies the problem by fixing $\alpha = 1$, resulting in computational efficiency and training stability.
> RL tasks are resource-intensive, and simultaneous optimization of $\alpha$ and $\beta$ introduces additional complexity and potential instability.
>
> While fixing $\alpha$ reduces flexibility, $\beta$ still captures kurtosis and indirectly controls variance.
> This trade-off is reasonable in practice, as $\beta$ alone provides sufficient expressiveness for most RL scenarios, particularly those with heavy-tailed distributions.
> In addition, since $\alpha$ and $\beta$ are interdependent, e.g., variance $\sigma^2$ depends on both $\alpha$ and $\beta$, joint optimization may result in parameter updates that are inconsistent with each other, leading to poor uncertainty modeling.
> Our experiments in Figures 15 empirically support the choice of fixing $\alpha$.
> It is worth noting that the practical impact from fixing $\alpha$ is minimal, as $\beta$ still adapts dynamically to changes in TD-error distributions.

---

> ### Author Response · Authors · 2024-11-25
>
> ## On Higher-Order Moments and Task Characteristics
>
> > Related to Question 7
>
> Thank you for this thoughtful question.
> Below, we address how the method's performance relates to the characteristics of the TD-error distribution, including the role of higher-order moments, and clarify whether the method demonstrates greater improvements in environments with more pronounced or subdued higher-order moments.
>
> The proposed method leverages the shape parameter $\beta$ in the GGD to explicitly model higher-order moments, such as kurtosis.
> This capability enables the method to effectively handle TD-error distributions with heavy tails (high kurtosis), outperforming Gaussian-based approaches that fail to account for extreme errors adequately.
> In environments with larger tails, i.e., leptokurtic, the method demonstrates significant performance improvements by explicitly addressing tail behavior.
> Unlike baseline Gaussian models, which often underestimate or ignore the impact of extreme errors, the proposed approach focuses on less spread-out samples, enhancing robustness to noisy or outlier TD errors.
> This is empirically evident in Figure 2, where environments such as Hopper-v4 and Ant-v4 exhibit heavy-tailed TD-error distributions and benefit significantly from the proposed method.
>
> In contrast, for smaller tails (platykurtic distributions), the advantages of modeling kurtosis are less pronounced.
> However, the method remains effective due to its dynamic weighting scheme, which adjusts to the lower kurtosis by de-emphasizing higher-order moment terms when they are less relevant.
> This adaptability prevents overfitting to low-noise regions, a common limitation of variance-based baselines.
> As a result, the proposed method maintains robustness by continuously adapting to the evolving characteristics of the TD-error distribution.

---

> > ### Comment · Reviewer_W9ns · 2024-11-28
> >
> > Thanks to the authors for the thorough answer and classification. The extra background on equation 3 is very useful, I understand the details much better, and most of my concerns where addressed. Having said this, I am still confused about a few points:
> >
> > The answer to Q6 is quite reasonable, I understand that optimizing $\alpha$ and $\beta$ simultaneously might lead to inconsistent parameter pairs, perhaps one way to solve this is by re-projecting to a valid pair. I still do not understand why estimating only $\beta$ is sufficient. Perhaps a good way to clarify this would be to demonstrate which pairs of variance and kurtosis $\beta$ can describe for a fixed $\alpha$, and show that the these moments from the TD-errors empirical distributions overlap with these pairs. If using the generalized Gaussian distribution (GGD) better describes the distributions of TD-errors by tuning a single parameter $\beta$, then the pairs of empirical second and fourth moments from the TD-error distribution should overlap with the pairs described by varying $\beta$ (as variance and kurtosis are coupled for a fixed $\alpha$ and $\beta$). Also, what are the distributions where $\beta$ alone is insufficient?
> >
> > The question above is also related to Q7. In state trajectories where rewards are frequent, the TD-error will correspond to a large sum of rewards and could become Gaussian. Is adjusting $\beta$ of a GGD sufficient in such cases? My guess is that you would need to normalize the TD-errors in some way, e.g. scaling the empirical distribution to have unit variance, to fit a proper $\beta$. Otherwise, you would essentially be fitting two numbers (variance and kurtosis) using only one parameter ($\beta$). Presumably there is a detail I am missing here.
> >
> > Thank you.

---

> > > ### Author Response · Authors · 2024-12-03
> > >
> > > Thank you for your thoughtful feedback and for recognizing the improvements.
> > > We appreciate the opportunity to address the remaining concerns and provide additional contexts.
> > >
> > > **Sufficiency and Limitation of $\beta$ for GGD Modeling.**
> > > We agree that visualizing the variance-kurtosis pairs describable by the GGD for a fixed $\alpha$, and comparing them with empirical TD error distribution would provide valuable insights.
> > > As noted in Section 3.1 of the revised paper, the variance $\sigma^2$ and kurtosis $\kappa$ for a fixed $\alpha$ depend on $\beta$ as follows:
> > >
> > > $$
> > >   \sigma^2 = \frac{\Gamma(3/\beta)}{\Gamma(1/\beta)}, \kappa = \frac{\Gamma(5/\beta)\Gamma(1/\beta)}{\Gamma(3/\beta)^2}-3.
> > > $$
> > >
> > > These equations reveal an intrinsic coupling, where changes in $\beta$ simultaneously affect both variance and kurtosis.
> > > As $\beta$ decreases, both variance and kurtosis increase, aligning with the observed relationship demonstrated in Figure 1 of [the appended document](https://drive.google.com/file/d/1IV-HGTpbYyEMwrXLzfz-btJCDYy9Uv_g).
> > > Since high variance is often a reflection of larger outliers or extreme values, these extreme values disproportionately influence higher-order moments _for heavy-tailed distributions like TD error distributions_, as seen in the definition of kurtosis:
> > >
> > > $$
> > >   \kappa = \frac{E[(X-\mu)^4]}{E[(X-\mu)^2]^2}.
> > > $$
> > >
> > > In such cases, the larger magnitude of the numerator highlights the sensitivity of kurtosis to outliers, reinforcing the relationship between variance and kurtosis in heavy-tailed distributions.
> > >
> > > As it is practically difficult to compute the empirical kurtosis-variance pairs of TD errors, we compared the empirical TD error distributions with GGDs parameterized by only $\beta$ (from model) and those optimized for both $\alpha$ and $\beta$ (using SciPy), to validate whether $\beta$ alone is sufficient.
> > > Figure 2 of [the appended document](https://drive.google.com/file/d/1IV-HGTpbYyEMwrXLzfz-btJCDYy9Uv_g) reveals that using only $\beta$ closely matches the empirical distribution, providing strong evidence for the effectiveness of this approach and the sufficiency of $\beta$ in capturing variance and kurtosis in general cases.
> > > Notably, as training progresses and $\beta$ estimation gets accurate, the GGDs closely approximate the empirical distributions, supporting the choice of $\beta$ as the sole parameter.
> > >
> > > While estimating $\beta$ has proven sufficient for the tested environments, there are potential distributions that may fall outside the describable range of GGDs with fixed $\alpha$.
> > > High variance but low kurtosis (platykurtic) or low variance but high kurtosis (leptokurtic) distributions may not align well with the GGD’s coupled moments, along with multi-modal distributions.
> > > While such cases were not observed in our experiments, we propose exploring joint optimization of $\alpha$ and $\beta$ or alternative models as future work to address these cases.
> > >
> > > **Gaussianity in TD Error Distributions with Frequent Rewards.**
> > > Regarding the possibility of Gaussian TD errors in trajectories with frequent rewards, we emphasize that the GGD inherently converges to a Gaussian as $\beta\to2$.
> > > This flexibility allows our model to adapt to such scenarios without any additional modifications.
> > >
> > > Additionally, while our current method does not explicitly normalize TD errors, the risk-averse weighting implicitly accounts for their scale, ensuring robustness across varying distributions.
> > > Nonetheless, normalizing TD errors, e.g., scaling to unit variance, could further improve numerical stability and facilitate consistent $\beta$ fitting.
> > > This is an intriguing direction for future work, especially in environments with highly dynamic reward scales.

---

### Official Review · Reviewer_Nr5G · 2024-11-04

**Soundness:** 3
**Presentation:** 2
**Contribution:** 3
**Rating:** 6
**Confidence:** 3

**Summary:**

The paper presents a novel framework for generalized Gaussian error modeling in uncertainty-aware temporal difference (TD) learning. It critiques conventional methods that assume a zero-mean Gaussian distribution for TD errors, leading to inaccurate uncertainty estimations. The proposed framework incorporates higher-order moments, specifically kurtosis, to enhance error modeling in reinforcement learning.

**Strengths:**

1. The paper offers a closed-form expression demonstrating the relationship between uncertainty and the generalized Gaussian distribution's shape parameter, adding depth to the theoretical framework.
2. The framework's applicability to both discrete and continuous control settings makes it relevant across various reinforcement learning contexts.
3. The emphasis on data-dependent noise and aleatoric uncertainty is timely and important for improving the robustness of reinforcement learning algorithms.

**Weaknesses:**

1. The introduction of higher-order moments may complicate the implementation in practical scenarios, which could deter application by practitioners.
2. In Figure 5, the return curves of Ant, HalfCheetah, and Humanoid are still increasing at the end of training step. The training steps can be increased to compare the final performances when all algorithms are converged.

**Questions:**

What specific tasks or environments in the real world do the authors envision as most beneficial for applying their method?

---

> ### Author Response · Authors · 2024-11-25
>
> ## On the implementation
>
> > Related to Weakness 1
>
> We appreciate the reviewer’s concern regarding the complexity of incorporating higher-order moments.
> To mitigate this challenge, our method is designed with simplicity and practicality in mind.
>
> As mentioned in Remark 2, we only employ $\beta$ estimation for GGD error modeling, setting $\alpha=0$, to minimize computational overhead while preserving the expressiveness needed to capture kurtosis.
> Such simplification ensures the implementation aligns closely with existing variance-based methods, requiring minimal modifications to architecture or training pipelines.
> Additionally, empirical results, as shown in Figures 4, 8, and 10, demonstrate both coefficients of variation (CV) and estimated $\beta$ values converge stably, indicating the stable training process of our method.
>
> ## On Task Applicability
>
> > Related to Question
>
> Thank you for raising this important question regarding real-world applications.
> The proposed approach is particularly well-suited for tasks and environments requiring robust decision-making under uncertainty and non-Gaussian error distributions.
> Below, we provide examples of such scenarios from two key application areas:
>
> 1. Robotics:
>    Tasks such as robot control, navigation, and manipulation in dynamic or unstructured environments often encounter:
>
>    - Aleatoric Uncertainty:
>      Stochastic dynamics, sensor noise, and imperfect actuators generate heavy-tailed error distributions.
>    - Epistemic Uncertainty:
>      Unexplored regions of the state space contribute to uncertainty during training.
>
>    By dynamically modeling kurtosis, our method improves policy robustness, enabling robots to handle stochastic and uncertain scenarios more effectively.
>
> 2. Finance:
>    Reinforcement learning applications in portfolio optimization and trading strategies involves:
>
>    - Heavy-tailed Distributions:
>      Financial returns and risk metrics often exhibit non-normal characteristics due to rare but impactful market events.
>
>    - Uncertainty:
>      Limited historical data and scarce market information introduce both aleatoric and epistemic uncertainties.
>
>    By modeling the evolving tailedness of TD errors, our framework enhances risk-sensitive strategies, enabling better adaptation to volatile market conditions while managing risk effectively.

---

### Official Review · Reviewer_UVhA · 2024-11-07

**Soundness:** 3
**Presentation:** 3
**Contribution:** 2
**Rating:** 6
**Confidence:** 3

**Summary:**

The authors consider uncertainty estimation in TD learning and seek to generalize the conceptually simple and implicit Gaussian assumption behind the MSE loss. To do so, they consider  the generalized gaussian distribution, which has an additional shape parameter that can modulate heavy/light tailedness for the errors. The authors extend a previous approach to estimating uncertainty in TD learning to include an additional network to predict the kurtosis/shape parameter.

Edit: updated my score after reviewing the author replies.

**Strengths:**

- Effective uncertainty estimation is an important problem for RL and the authors study a valid relaxation of commonly held simplifying assumptions around Gaussianity.
- The authors conduct an experimental study that shows plausible benefits of the proposed generalizations in simple environments

**Weaknesses:**

- There appear to be several ways to generalize the Normal distribution to have different tail behaviors, for example see q-Gaussian for yet another alternative. In the tradeoff between simplicity and more expressive modeling, it is unclear what is the right axis to explore. Even more generally, all of these are unimodal, symmetric (i.e. zero skew) distributions and arguably one could consider a full return distribution as well. In fact, prior work on distributional RL has explored this (https://arxiv.org/abs/1707.06887).

- In terms of the learning algorithm, the paper modifies the loss function proposed in [Mai et al 2022] to the GGD case aided with an additional predictor for the kurtosis (the beta term).

- The empirical results are mostly within error bars of the baselines. This seems especially so, when considering the marginal benefits in going from a basic uncertainty modeling with the gaussian assumption to the extra parameter estimation. Considering that this involves a whole extra network (not just an extra hyper-parameter) to predict the beta, this seems like a much more complex method for relatively little gain.

- The tailedness of the distribution seems like a very sensitive parameter to estimate, and sensitive to simple reward transformations and/or clipping so I would be surprised if these observations are robust to such changes.

**Questions:**

- It seems like the distribution associated with environment transition stochasticity could easily lead to multi modal distributions of the cumulative return, which might be a much more interesting/important aspect than estimating the shape of a unimodal distribution better. Please address whether you observed any evidence of multimodality in your empirical results, and if so, how your method handles or could be extended to handle such cases.

- Rewards are typically bounded, so I would expect any estimation of the tail behavior to be quite sensitive to various practical assumptions. Please describe any preprocessing steps applied to the rewards, and how the performance varies with different reward scales or bounds.

- Given that your method requires predicting an extra head for the beta/kurtosis parameter, and training this (see Equation (4)) requires gradient descent through the gamma function of the output, how stable is the learning and/or optimization?  Please consider providing empirical evidence of the optimization stability, such as plots of the beta parameter estimates over time, or  discuss any specific techniques used to ensure stable training.

---

> ### Author Response · Authors · 2024-11-25
>
> ## On the Contribution of the Paper
>
> > Related to Weaknesses 2 and 3
>
> We acknowledge the reviewer's concern about the novelty of our work and the effectiveness of the proposed method.
>
> Our work is motivated by the limitations of conventional temporal difference (TD) learning methods that assume a zero-mean Gaussian distribution for TD errors.
> Specifically focusing on the shape of the TD error distribution, we propose a novel uncertainty-aware objective function that minimizes the negative log-likelihood of the generalized Gaussian distribution (GGD) of the TD errors.
> From the existing literature on uncertainty-aware reinforcement learning (RL), which utilizes variance head for uncertainty estimation ([Kendall & Gal, 2017](https://arxiv.org/abs/1703.04977), [Mai et al., 2022](https://arxiv.org/abs/2201.01666)), we extend the method to include higher-order moments, specifically kurtosis, to enhance the description of the TD error distribution.
>
> We investigate this extension both empirically and theoretically, providing insights into the effects of higher-order moments on the TD error distribution.
> Based on those results, implications of our work to mitigate both aleatoric and epistemic uncertainties in TD learning lead to improved performance across various settings.
>
> Contrary to concerns about increased complexity, our method requires only a single additional network head to estimate $\beta$, as conventional variance estimation requires.
> This makes the proposed method straightforward to integrate into existing architectures.
> We believe this simplicity is a strength of our method, as it allows practitioners to easily extend their uncertainty-aware TD learning methods to include higher-order moments, without significant changes to the existing framework.
>
> ## On the Choice of the Generalized Gaussian Distribution
>
> > Related to Weakness 1 and Question 1
>
> We appreciate the reviewer's comment on the choice of the GGD and the potential for exploring other distributions.
>
> The GGD was selected due to its flexibility in capturing varying tail behaviors through the shape parameter $\beta$.
> This aligns well with the observed heavy-tailed distributions in TD errors, as shown in Figures 2, 6, and 7.
> While alternatives such as $q$-Gaussian distributions could also be considered, they introduce similar parameter estimation challenges without offering significant advantages over the GGD.
>
> Methods like particle-based distributional RL ([Bellemare et al., 2017](https://arxiv.org/abs/1707.06887), [Nguyen et al., 2020](https://arxiv.org/abs/2007.12354)) model the entire return distribution, often focusing on its variance.
> Our work differs in emphasizing the shape of the TD error distribution, particularly its tailedness, which is crucial for understanding the uncertainty in TD learning.
> Furthermore, parametric approaches are computationally more efficient, making them more suitable for online learning in RL.
>
> We agree that incorporating skewness or handling multimodal distributions are also interesting extensions.
> However, our experimental results indicate that TD error distributions are unimodal in our setups, making GGD a practical and effective choice for this work.
> Additionally, as we hypothesized in Lines 273-276, exploration in RL can lead to heavy-tailed distributions, making the tailedness of the distribution more critical for understanding the uncertainty in TD learning than skewness.
>
> ## On Training Stability
>
> > Related to Question 3
>
> We try to show the stability of the training process by providing coefficients of variantion (CV) of the $\beta$ estimates over time in Figures 4 and 8.
> The CV of $\beta$ estimation are lower than variance estimation, which indicates greater stability in estimating the shape parameter of the GGD.
> Please note that the convergence of $\beta$ estimation, given in Figure 10, is also more stable than variance estimation.
>
> To further ensure stability, we employed the following techniques (Note the Implementation section in Appendix C.):
>
> 1. Normalized Parameter Range: We applied the softplus function, a smooth approximation to the ReLU function ([Dugas et al., 2000](https://papers.nips.cc/paper_files/paper/2000/file/44968aece94f667e4095002d140b5896-Paper.pdf)), to the outputs, to constrain $\beta$ is constrained within a range to avoid extreme values that could destabilize training
> 2. Simplified Loss Computation: We modify the NLL loss for the GGD by employing
>    $\mathbb{Q_\beta}$ as a multiplier rather than as an exponent, to reduce computational overhead and mitigate numerical instability.
>
> These measures, combined with empirical evidence provided in the paper, support the robustness and stability of the optimization process.

---

> ### Author Response · Authors · 2024-11-25
>
> ## On Sensitivity to Reward Scales and Bounds
>
> > Related to Weakness 4 and Question 2
>
> Our method focuses on the distribution of TD errors, which are not directly related to the reward scale.
> This is because estimation of TD errors is less biased than the reward ([Flennerhag et al., 2020](https://arxiv.org/abs/2010.02255)).
> Therefore, the sensitivity of the parameter estimation to reward scales is less of a concern in our method.

---

### Author Response · Authors · 2024-11-25

We thank the reviewers for their insightful comments and questions, which have significantly helped us improve the manuscript.
In response to the feedback, we have made the following revisions:

1. Enhanced the description of prior work in Section 2.1 to provide clearer context.
2. Detailed the role of ensembled critics in uncertainty estimation. These updates are included in Sections 2.1 and 3.2..
3. Defined "unexpected" states and rewards and elaborated on the exploratory hypothesis in Section 3.1.1.
4. Provided a more explicit connection between Theorem 2 and the risk-averse weighting mechanism in Section 3.1.2.
5. Addressed other minor comments to improve clarity, consistency, and presentation throughout the manuscript.

These updates are marked in blue in the revised manuscript for easy reference.
We hope the reviewers find these revisions satisfactory and appreciate their valuable input in refining our work.

---

### Author Response · Authors · 2024-12-04

Dear Reviewers,

We hope that you have had an opportunity to review our responses and clarifications provided during the discussion period.
As the deadline for the rebuttal phase approaches, we kindly request confirmation on whether our updates have sufficiently addressed your concerns.

Thank you again for your time and thoughtful review.

Best regards,
The Authors

---

### Meta-Review · Area_Chair_VvdB · 2024-12-29

**Metareview:**

This paper addresses the limitations of conventional uncertainty-aware Temporal Difference (TD) learning methods, and proposes a more sophisticated and flexible way to model uncertainty in TD learning by using the GGD and explicitly considering higher-order moments, leading to improved performance in policy gradient algorithms.

One of the major contribution from the paper, i.e., the loss function design, is modified from [Mai et al 2022] with the GGD case aided with an additional predictor for the kurtosis (the beta term). Mwanehile, all the theoretial properties are stemmed from prior work. Moreover, there are several alternative ways to generalize the Normal distribution to have different tail behaviors, especially the distributional RL. However, there is no empirical comparison conducted, which makes the significance of the paper difficult to be justified.

In sum, this paper is promising, however, the current version is not ready to be published. I encourage the authors to take the reviewers' suggestions into account to improve the paper.

**Additional Comments On Reviewer Discussion:**

This is a borderline paper. The authors addressed several concerns raised by reviewers. However, the novelty and significance of the work, raised by almost every reviewer, has not been clearly revealed.

---

### Decision · Program_Chairs · 2025-01-22

Reject